# Large haploblocks underlie rapid adaptation in the invasive weed *Ambrosia artemisiifolia*

Paul Battlay[1,10], Jonathan Wilson [1,10], Vanessa C. Bieker [2], Christopher Lee[1], Diana Prapas[1], Bent Petersen[3,4], Sam Craig [1], Lotte van Boheemen[1], Romain Scalone [5,6], Nissanka P. de Silva[1], Amit Sharma[7], Bojan Konstantinović[8], Kristin A. Nurkowski[1,9], Loren H. Rieseberg[9], Tim Connallon[1], Michael D. Martin [2,11] & Kathryn A. Hodgins [1,11] ✉

Adaptation is the central feature and leading explanation for the evolutionary diversification of life. Adaptation is also notoriously difficult to study in nature, owing to its complexity and logistically prohibitive timescale. Here, we leverage extensive contemporary and historical collections of *Ambrosia artemisiifolia*—an aggressively invasive weed and primary cause of pollen-induced hayfever—to track the phenotypic and genetic causes of recent local adaptation across its native and invasive ranges in North America and Europe, respectively. Large haploblocks—indicative of chromosomal inversions—contain a disproportionate share (26%) of genomic regions conferring parallel adaptation to local climates between ranges, are associated with rapidly adapting traits, and exhibit dramatic frequency shifts over space and time. These results highlight the importance of large-effect standing variants in rapid adaptation, which have been critical to *A. artemisiifolia*'s global spread across vast climatic gradients.

Adaptation can be rapid, yet changes in the traits and genes that underlie adaptation are difficult to observe in real time because speed is relative: that which is fast in evolutionary terms is slow from the human perspective. Thus, while we know that adaptation is central to evolutionary diversification, species persistence, and invasiveness, the genetic and phenotypic dynamics of adaptation are difficult to document outside of the laboratory.

Invasive species are powerful systems for characterizing adaptation in nature, owing to several features that make them unique. In particular, biological invasions coincide with exceptionally rapid evolution[1–3], observable over human lifespans, as invasive populations can encounter drastically different environmental conditions from those of their source populations. Many invasions are, moreover, documented in large, geo-referenced herbarium collections, which can be phenotyped and sequenced to identify and track adaptive evolutionary changes through time—feats that are rarely achieved in natural populations. Invasive species frequently inhabit broad and climatically diverse ranges, which favors the evolution of adaptations to local environmental conditions[2,3], along with evolved tolerance of environmental extremes, which may be conducive to invasiveness[4].

[1]School of Biological Sciences, Monash University, Melbourne, Victoria, Australia. [2]Department of Natural History, NTNU University Museum, Norwegian University of Science and Technology (NTNU), Trondheim, Norway. [3]Center for Evolutionary Hologenomics, GLOBE Institute, University of Copenhagen, Copenhagen, Denmark. [4]Centre of Excellence for Omics-Driven Computational Biodiscovery (COMBio), AIMST University, 08100 Bedong, Kedah, Malaysia. [5]Department of Crop Production Ecology, Uppsala Ecology Center, Swedish University of Agricultural Sciences, Uppsala, Sweden. [6]Department of Grapevine Breeding, Hochschule Geisenheim University, Geisenheim, Germany. [7]Cell, Molecular Biology and Genomics Group, Department of Biology, Norwegian University of Science and Technology (NTNU), Trondheim, Norway. [8]Department of Environmental and Plant Protection, Faculty of Agriculture, University of Novi Sad, Novi Sad, Serbia. [9]Department of Botany and Biodiversity Research Centre, University of British Columbia, Vancouver, Canada. [10]These authors contributed equally: Paul Battlay, Jonathan Wilson. [11]These authors jointly supervised this work: Michael D. Martin, Kathryn A. Hodgins. ✉e-mail: kathryn.hodgins@monash.edu

Because they often occupy geographically broad native and invasive ranges, invasive species allow for tests of the predictability of evolution—a major puzzle in biology—as local adaptation across native and invasive ranges may favor either parallel or unique genetic solutions to shared environmental challenges. However, despite the promise of historical records and other features of invasive species that make them tractable systems for capturing adaptation in action, this treasure trove of data has not been fully utilized to elucidate the genetic basis of local adaptation during recent range expansions.

*Ambrosia artemisiifolia*, an annual weed native to North America, has mounted successful invasions on all continents except Antarctica[5]. The species thrives in disturbed habitats and has experienced extensive range shifts, historically documented in pollen records and herbarium collections. It also produces highly allergenic pollen, which is the chief cause of seasonal allergic rhinitis and asthma in the United States[6]. In Europe, approximately 13.5 million people suffer from *Ambrosia*-induced allergies, costing ~7.4 billion euros annually[7]. Continued invasion and climate change are predicted to more than double sensitization to *Ambrosia* pollen[8], further magnifying the health burden of this pest. Pollen monitoring has demonstrated that climate change has already significantly lengthened the ragweed pollen season, particularly at high latitudes[9]. Consequently, there is considerable incentive to understand the factors that contribute to *Ambrosia* pollen production, including the species' invasive spread, the timing of pollen production, plant size, and fecundity.

*Ambrosia artemisiifolia* populations are characterized by strong local adaptation and high gene flow between populations[10,11]. In Europe, invasive populations have been established through multiple genetically diverse introductions from North America over the last ~160 years[11–13]. Remarkably, latitudinal clines observed for multiple traits in the native range, including flowering time and size, have re-evolved in Europe and Australia[14], suggesting rapid local adaptation following invasion. Moreover, this trait-level parallelism is echoed in signals of parallelism at the genetic level[10].

As biological invasions continue to increase worldwide[15] and the effects of anthropogenic climate change intensify, understanding the genetic architecture of adaptation to sudden environmental shifts—a classical question in evolutionary research—becomes ever more important. While long-standing theory suggests that evolution in response to incremental environmental change should proceed through mutations of infinitesimally small[16] or moderate[17] effect, large-effect mutations are predicted to be useful in bridging extreme, sudden environmental shifts[18]. Moreover, alleles of large effect will, in cases of local adaptation, be better able to persist in the face of the swamping effects of gene flow[19]. Large-effect mutations are also more likely to produce patterns of evolutionary repeatability, or genetic parallelism, between species' ranges[20], particularly if adaptive responses make use of standing genetic variation (as would be expected during a bout of rapid adaptation), rather than de novo mutations[21]. These features of large-effect mutations may, however, be achieved by groups of mutations in tight genetic linkage[19], including mutations captured by chromosomal inversions[22]. There is substantial empirical evidence for the involvement of inversions in local adaptation[23]. For example, *Drosophila melanogaster*'s *In(3R)Payne* inversion shows parallel environmental associations across multiple continents[24], and several plant inversions have been identified as contributing to local adaptation and ecotype divergence[25,26]. Theory also predicts that inversions can drive range expansions[27], though their actual contributions to biological invasions are not well-understood.

Here, we develop a chromosome-level phased diploid reference assembly, and examine genome-wide variation in over 600 modern and historic *A. artemisiifolia* samples from throughout North America and Europe[11]. Using these data of unparalleled spatial and temporal resolution, we first identify regions of the genome experiencing climate-mediated selection in the native North American and introduced European ranges leveraging landscape genomic approaches and genome-wide association studies of adapting traits such as flowering time. Second, motivated by evidence that European and North American populations show similar trait clines with respect to climate[14], we examine the extent of between-range parallelism at the genetic level. Although adaptive traits such as flowering time and size are polygenic, we expect to see substantial levels of parallelism if large and moderate effect standing variants were contributing to adaptive divergence. Third, we determine if these same regions show temporal signatures of selection in Europe, which would be expected if some invading populations were initially maladapted to their local climates. We couple this temporal genomic analysis with a temporal analysis of phenological trait changes in European herbarium samples to further support our findings of genomic signatures of selection on flower time genes. Finally, we identify haplotype blocks with multiple features consistent with inversions, in genomic regions enriched for signatures of parallel adaptation. Four of these colocalize with inversions identified in our diploid assembly, confirming their identity as inversions. To determine if these haploblocks contribute to rapid local adaptation in Europe, we assess spatial and temporal changes in their frequency as well as their associations with adaptive traits.

## Results
### Reference genome assembly
We assembled a chromosome-level phased diploid *Ambrosia artemisiifolia* reference genome (Fig. 1; Supplementary Fig. 1) from a heterozygous, diploid individual collected from Novi Sad, Serbia. After scaffolding with HiRise[28], our final assembly consisted of two haploid assemblies with genome sizes of 1.11 and 1.07 Gbp (flow cytometry estimates of genome size range from 1.13–1.16 Gbp[29,30]), with 94.3 and 96.5% of each respective genome assembled into 18 large scaffolds (Supplementary Table 1; Supplementary Table 2), consistent with the 18 chromosomes of *A. artemisiifolia*. Complete copies of all 255 Benchmarking Universal Single-Copy Orthologs (BUSCO[31]) genes were identified on the 18 chromosomes of each haploid assembly, with 183 (71.8%) single-copy and 72 (28.2%) duplicated (Fig. 1C; Supplementary Table 1). These high numbers of duplicated BUSCO genes likely reflect the whole-genome duplications experienced in the Asteraceae, including a recent one shared by *Helianthus* (sunflower) species at the base of the tribe[32,33]. This species also retained a large number of duplicated BUSCO genes[33]. A large fraction of the genome consisted of repetitive sequence (67%; Supplementary Data 1). Retroelements were the largest class (39.5%), with long terminal repeats, particularly Gypsy (7.87%) and Copia (18.98%), being the most prevalent. MAKER[34] identified 36,826 gene models with strong protein or transcript support, with average coding lengths of 3 kbp and 5.75 exons per gene (Fig. 1A; Supplementary Table 3).

### Genome-wide association studies
Genome-wide association studies (GWAS) using 121 modern samples across North American ($n = 43$) and European ($n = 78$; Supplementary Data 2) ranges identified significant associations with 16 of 30 phenotypes, many of which are putatively adaptive, previously measured by van Boheemen, Atwater and Hodgins[14] (Supplementary Fig. 2; Supplementary Data 3). All phenotypes yielded associations within at least one predicted gene, including an association between three flowering time phenotypes and a non-synonymous SNP in the *A. artemisiifolia* homolog of *A. thaliana* flowering-time pathway gene *early flowering 3* (*ELF3*)[35], an evolutionary hotspot for parallel flowering time adaptation in *A. thaliana*, barley and rice[36] (Fig. 2A, B; Supplementary Data 3). Candidate SNPs in *ELF3* are restricted to high-latitude populations in both ranges, where they occur at moderate to high frequencies (Fig. 2C). While the latitudes of these populations are greater in Europe than North America, the climatic conditions are similar (Supplementary Fig. 3), indicative of local climate adaptation in

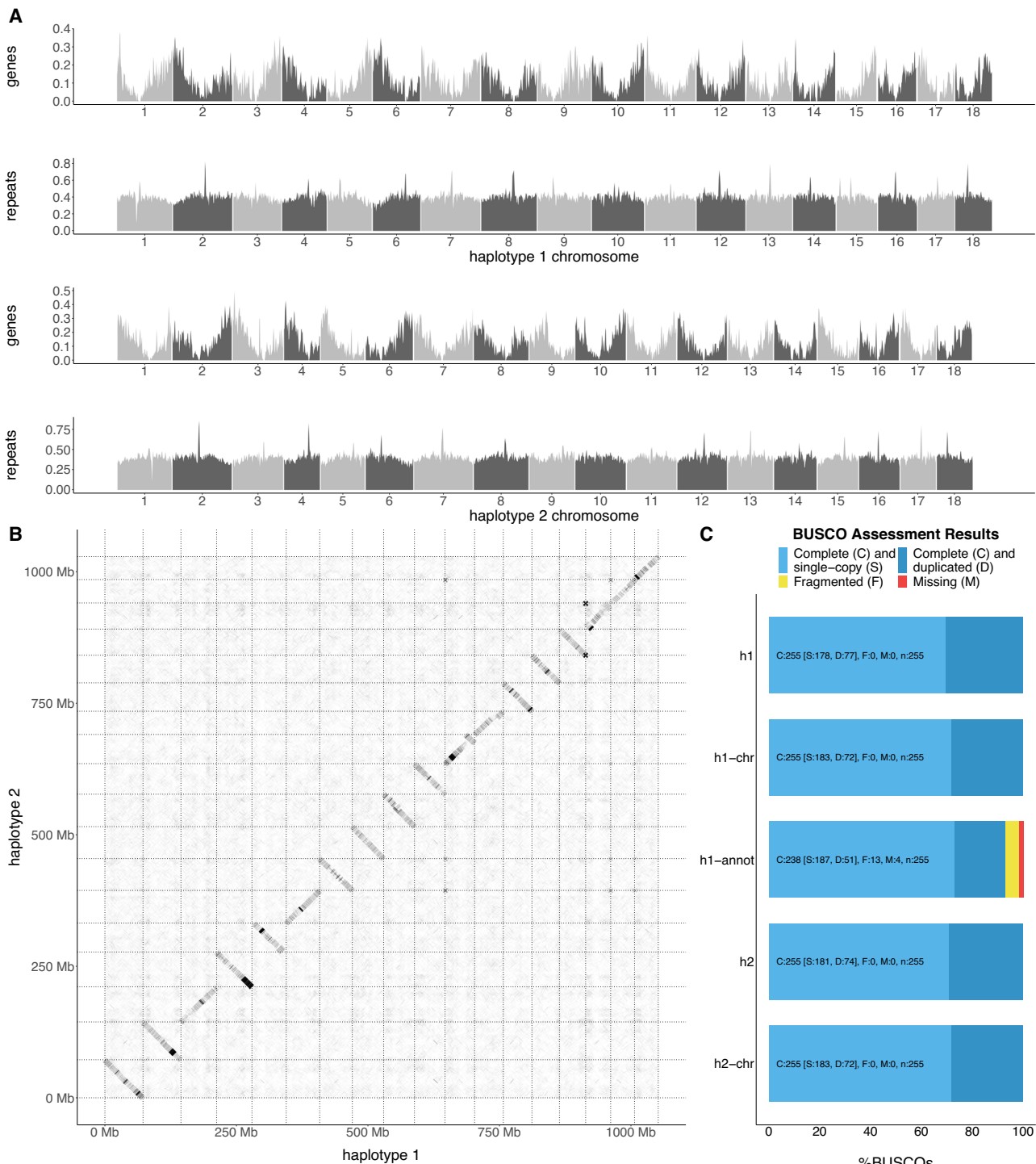

**Fig. 1 | A phased diploid genome assembly of *Ambrosia artemisiifolia*. A** The distribution of gene and repeat density in 1 Mbp windows across the 18 chromosomes of each haplotype. **B** Alignments of the 18 chromosomes of each haplotype. **C** Benchmarking Universal Single-Copy Orthologs (BUSCO) results for each haplotype assembly (h1; h2), each chromosome-only assembly (h1-chr; h2-chr), and the gene annotations for haplotype 1 (h1-annot) using the eukaryota *odb10* gene set. Source data are provided as a Source data file.

parallel between ranges. In addition, a haplotype containing four non-synonymous SNPs in an S-locus lectin protein kinase gene was associated with several traits including flowering end date and maximum height (Supplementary Fig. 2; Supplementary Data 3).

## Environmental-allele associations

To identify genome-wide spatial signatures of local adaptation in North American and European ranges, we performed genome scans for population allele frequencies among *A. artemisiifolia* modern samples that were both highly divergent between populations (BayPass XtX)[37], and correlated with 19 WorldClim temperature and precipitation variables (Supplementary Table 4)[38]. Statistics were analyzed in 10 kbp windows using the Weighted-Z Analysis (WZA)[39]. In North America (143 samples; 43 populations), 2167 (80.1%) of the 2704 outlier windows for genomic divergence (XtX) were also outlier windows for at least one environmental variable (XtX-EAA),

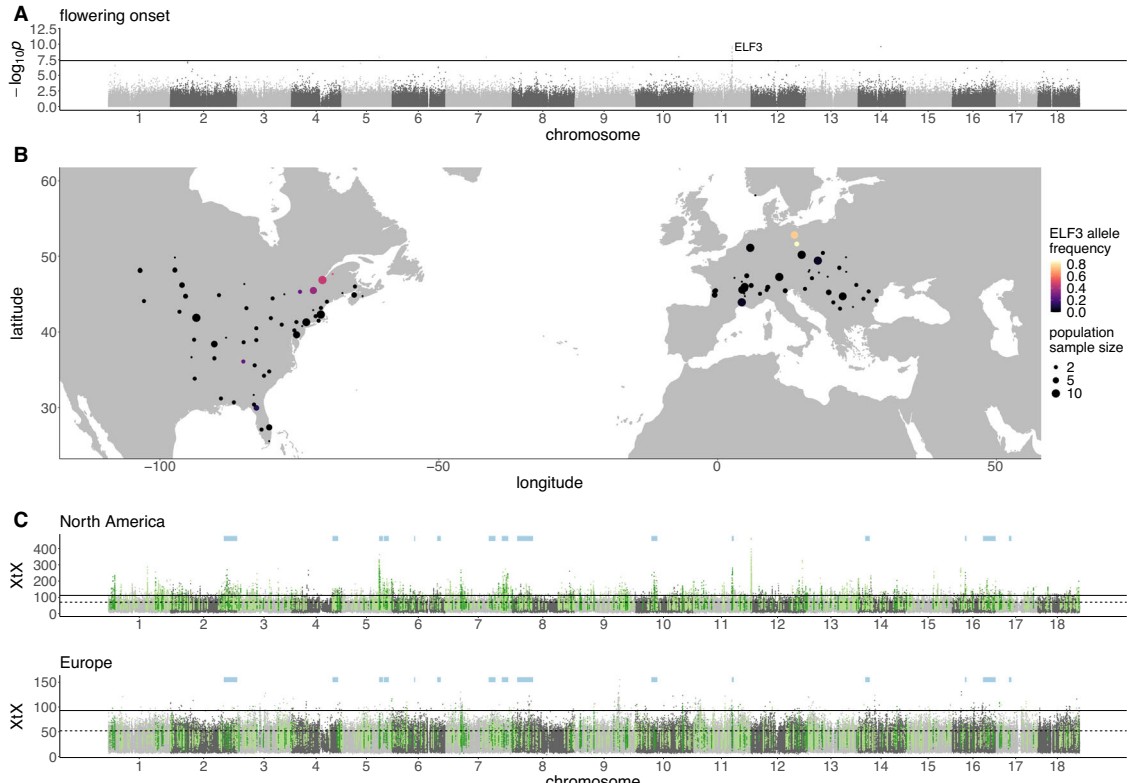

**Fig. 2 | Signatures of climate adaptation in *Ambrosia artemisiifolia*. A** GWAS $-\log_{10}p$-value of mixed model association against genomic location for flowering onset (solid line indicates a Bonferroni-corrected *p*-value of 0.05). **B** Distribution of a strongly-associated non-synonymous SNP in *ELF3* among modern *A. artemisiifolia* populations used in this study. **C** Genome-wide XtX scans between sampling locations within each range separately. Solid lines indicate Bonferroni-corrected significance derived from XtX *p*-values; dashed lines indicate the top 1% of genome-wide XtX values. Green highlights represent the top 5% of 10 kbp WZA windows for each scan that are also among the top 5% of EAA WZA windows for at least one environmental variable, with dark green indicating outlier windows shared between North America and Europe. Pale blue bars indicate the location of 15 haploblocks (putative chromosomal inversions) that overlap shared outlier windows. Source data are provided as a Source data file.

while in Europe (141 samples; 31 populations), only 1357 (50.3%) of the 2697 XtX outlier windows overlapped environmental variable outlier windows. Signatures of local adaptation were much stronger in North America than Europe, with the North American range showing more extreme XtX values (Fig. 2C), as well as more XtX-EAA windows (Fig. 2C; Supplementary Table 4). This suggests that North American *A. artemisiifolia* exhibits greater population differentiation, and a stronger relationship between population differentiation and the environment than Europe, which is consistent with the expectation that populations from the native range will be better-adapted to their environment than those from the recently-invaded European range.

Previous studies in *A. artemisiifolia* have identified signatures of repeatability between native and invasive ranges at phenotypic and genetic levels[10,14]. We observed congruent patterns in our data: among North American and European XtX-EAA outlier windows, 291 showed parallel associations with the same environmental variable between ranges (significantly more than would be expected by chance; hypergeometric $p = 1.07 \times 10^{-126}$; Fig. 2C), with 21.4% of climate adaptation candidates in Europe also candidates in North America. To account for the possibility that the number of parallel windows is inflated by extended linkage disequilibrium between windows (and hence represents a smaller number of loci), we combined consecutive outlier windows, and windows in haploblock regions, into single windows and repeated the analysis, in which the parallelism remained highly significant (hypergeometric $p = 1.29 \times 10^{-91}$). Consequently, many of the same regions of the genome are involved in climate adaptation in both ranges.

North American, European, and parallel XtX-EAA outlier windows included 28, 22, and three flowering-time pathway genes, respectively, however, this only represented a significant enrichment (Fisher's exact test $p < 0.05$) in North America (Supplementary Table 5). GWAS flowering time candidate *ELF3* was located in a parallel XtX-EAA window, while the flowering and height-associated S-locus lectin protein kinase gene was in a North American XtX-EAA window only. Gene ontology terms enriched in parallel XtX-EAA windows included "iron ion binding" and "heme binding" (terms relating to cytochrome P450 genes), as well as "gibberellin biosynthetic process" (Supplementary Data 4). Some cytochrome P450 genes are involved in detoxification of xenobiotic compounds and the synthesis of defense compounds, while others play key developmental roles[40], including contributing to the biosynthesis of gibberellin, a hormone that regulates a range of developmental events including flowering[41].

**Temporal phenotypic analysis**

To further identify the features of recent adaptation in *A. artemisiifolia*, we leveraged herbarium samples, collected from as early as 1830, for phenotypic and genomic analyses. A trait-based analysis of 985 digitized herbarium images identified a significant shift in the probability of flowering and fruiting over time in Europe, but this change depended on the latitude or the day of the year the sample was collected (Supplementary Fig. 4; Supplementary Table 6). For the trait presence of a mature male inflorescence, we identified a significant interaction between collection year and latitude ($F_{1886} = 7.89$, $p < 0.01$) where in northern populations, more recently collected plants were more likely to be flowering than older historic specimens. For this trait, collection

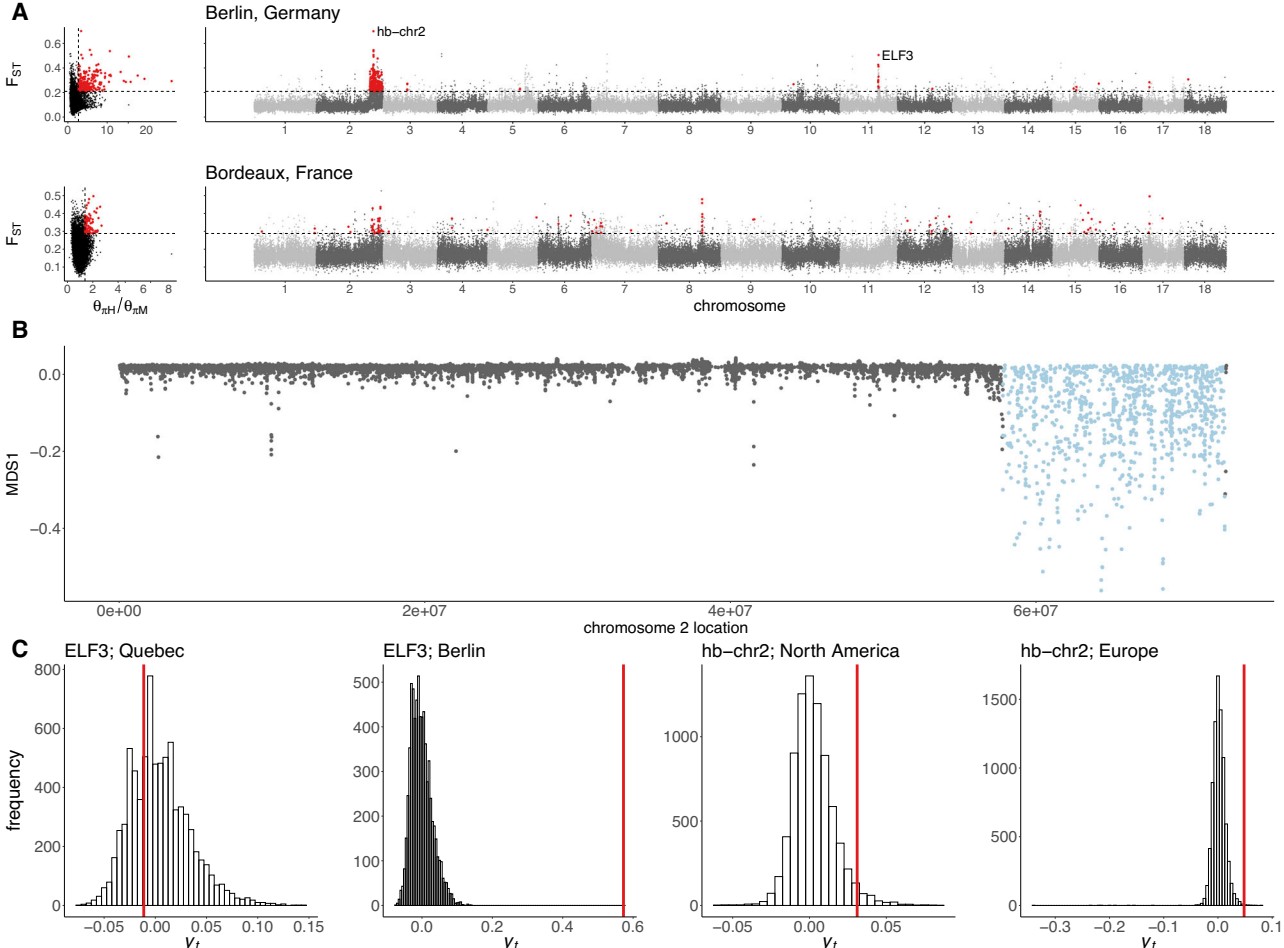

**Fig. 3 | Temporal signatures of selective sweeps in Europe. A** Distributions of $F_{ST}$ between historic and modern samples and the ratio of historic to modern nucleotide diversity ($\theta_{\pi H}/\theta_{\pi M}$) from Berlin and Bordeaux, and $F_{ST}$ against genomic location. Red points indicate putative selective sweep windows, which are in the top one percent of per-window $F_{ST}$ and $\theta_{\pi H}/\theta_{\pi M}$ (dashed lines). **B** Strong evidence for a selective sweep on chromosome 2 in European populations corresponds with local divergent population structure (MDS1), indicating the presence of a haploblock (putative chromosomal inversion; pale blue) in this region. **C** A standardized measure of allele frequency change, $y_t$ (calculated according to Eq. 1) for shifts between historic and modern populations across putatively neutral SNPs (histograms) and selective sweep candidates (red lines). Source data are provided as a Source data file.

day also significantly interacted with collection year and more recently collected plants were more likely to be flowering later in the year and less likely to be flowering earlier in the year. Similar patterns were identified with the presence of fruit, as older samples were less likely to produce fruit later in the season compared to recent samples (day-by-year interaction $F_{1886} = 32.33$, $p < 0.001$; Supplementary Fig. 4; Supplementary Table 7). This substantial spatio-temporal change in phenology is consistent with experimental common gardens that show that earlier flowering has evolved in northern populations and later flowering in southern populations following the invasion of Europe[14]. Further, this shift in both flowering and fruit set over time supports the hypothesis that an initial mismatch between the local environment and the genotypes present impacted the reproductive output of *A. artemisiifolia* during the early stages of colonization, particularly in northern Europe.

### Genomic signatures of local selective sweeps

Genome resequencing of *A. artemisiifolia* herbarium samples[11] allowed comparisons between historic and modern populations. We grouped historic samples based on their age and proximity to a modern population sample, resulting in five North American and seven European historic-modern population pairs (Supplementary Table 8), which were scanned for signatures of local selective sweeps by identifying windows with extreme shifts in allele frequency and extreme reductions in diversity over time. We found far more evidence for recent sweeps in Europe (476 unique windows) than in North America (129 unique windows; Supplementary Figs. 5 and 6; Supplementary Table 8), consistent with the expectation that a haphazardly-introduced invader will frequently be maladapted, initially, and undergo rapid adaptation to local environmental conditions following its introduction. The most dramatic selective sweep signature was observed in Berlin over a ~14 Mbp region on chromosome 2 (Fig. 3A; Supplementary Fig. 6), which accounts for 274 (58%) of the European sweep windows. In Berlin, sweep windows were also observed containing and surrounding the flowering onset GWAS peak that includes *ELF3*. Scans of Fay and Wu's *H* in modern populations provide further evidence for recent selection in these regions, and suggest more geographically widespread selection in the region on chromosome 2 (Supplementary Figs. 7 and 8). In comparisons of spatial and temporal signatures of selection, three and five sweep windows were also XtX-EAA outliers in North America and Europe, respectively. All such windows in Europe were in the chromosome 2 or *ELF3* regions. To further investigate the temporal shift associated with *ELF3*, we focused on the non-synonymous *ELF3* variant across all samples within a 200 km radius of Berlin (15 historic and eleven modern samples). The frequency of the variant increased from 2.6% to 73.9% between historic

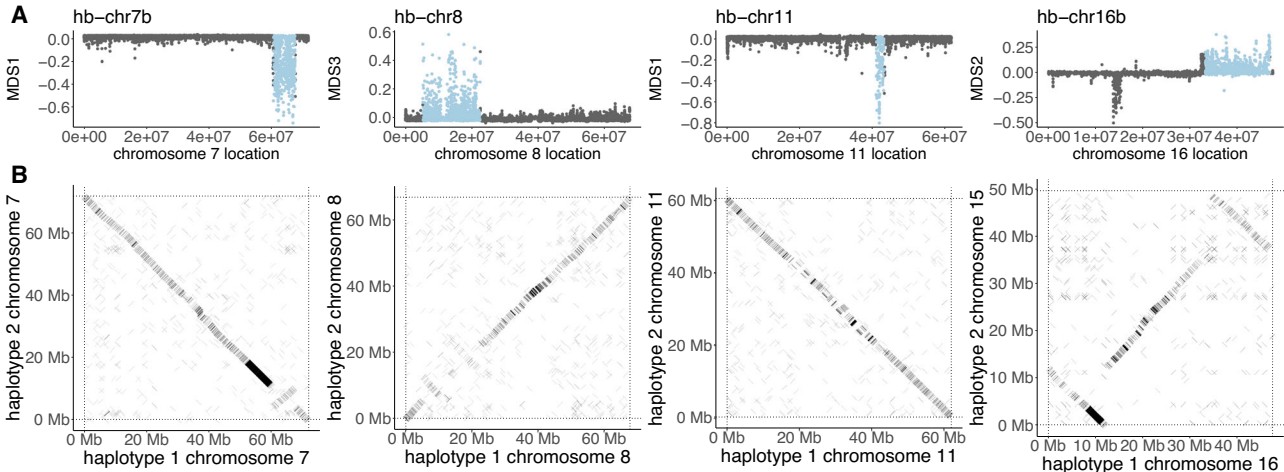

**Fig. 4 | Population-genomic signatures of inversions correspond to inversions segregating in the diploid reference. A** Divergent local population structure identifies putative inversions (pale blue regions) from population-genomic data. **B** Alignments of homologous chromosomes reveal inversion polymorphisms segregating in the diploid reference genome. Source data are provided as a Source data file.

and modern samples, an allele frequency shift greater than 10,000 putatively neutral loci sampled from the same geographic region; similar shifts were not observed in North American samples within a 200 km radius of Quebec City (ten historic and eleven modern), where the *ELF3* allele is at comparably high frequencies (Fig. 3C).

## Haploblock identification

Chromosomal inversions have previously been identified as driving local adaptation of ecotypes of *Helianthus* species[26]. We used a similar approach to identify genomic signatures of putative inversions (haploblocks) contributing to local adaptation in *A. artemisiifolia*. Briefly, we identified genomic regions in which population structure was divergent and fell into three clusters, putatively representing the heterozygous and two homozygous genotypic classes of an inversion. Further, we looked for pronounced shifts in population structure (indicating inversion breakpoints), elevated local heterozygosity in the heterozygous cluster, and increased linkage disequilibrium across the region (Supplementary Fig. 9). We examined mapping populations of *A. artemisiifolia*[42] for evidence of map-specific reductions in recombination across haploblock regions (i.e., suppressed recombination in haploblock regions in some maps but not others; Supplementary Fig. 10). This would be the pattern expected when recombination is suppressed by inversions in heterozygotes but not homozygotes, as opposed to the haploblocks being caused by global reductions in recombination in those regions. Most haploblocks with sufficient markers in the region showed evidence of suppressed recombination in some maps but not others, with the exception of hb-chr6b, which showed suppressed recombination in all maps. To validate our haploblock detection, we examined an alignment of our two haploid reference genomes and identified four segregating inversion polymorphisms corresponding to haploblocks (Fig. 4).

Focusing our analysis on regions showing signatures of adaptation, we identified 15 haploblocks with the above genomic signatures of chromosomal inversions overlapping the 291 WZA windows that were parallel outliers for both XtX and at least one climate variable: hb-chr2 (14.5 Mbp), hb-chr4 (6.2 Mbp), hb-chr5a (4.1 Mbp), hb-chr5b (5.4 Mbp), hb-chr6a (1.1 Mbp), hb-chr6b (3.9 Mbp), hb-chr7a (7.3 Mbp), hb-chr7b (7.0 Mbp), hb-chr8 (17.3 Mbp), hb-chr10 (6.5 Mbp), hb-chr11 (2.2 Mbp), hb-chr14 (4.9 Mbp), hb-chr16a (1.7 Mbp), hb-chr16b (13.7 Mbp) and hb-chr17 (2.8 Mbp). These haploblocks contained 77 of the 291 parallel XtX-EAA windows (26.5%; Fig. 2C), although they only represent ~10% of the genome (a significant enrichment; hypergeometric $p = 5.8 \times 10^{-17}$). One haploblock also corresponds to the

European selective sweep region on chromosome 2 (Fig. 3A, B). This suggests that these haploblock regions have played a pivotal role in generating parallel signatures of selection observed in *A. artemisiifolia*.

## Haploblock frequency changes through space and time

To identify changes in haploblock frequency over time and space, which would be consistent with selection on these putative inversions, we first estimated haploblock genotypes for all historic and modern samples. Within haploblock boundaries identified using modern sample SNP data in Lostruct[43], we performed local PCAs with both historic and modern samples (Supplementary Data 2) and identified genotypes by kmeans clustering (Supplementary Fig. 9). For modern samples, we used generalized linear models to estimate the slopes of the haploblock frequencies as a function of latitude within each range. For those haploblocks that were significantly associated with latitude, we compared these estimates with the genome-wide distribution of slopes for North America and Europe, based on 10,000 unlinked SNPs that were randomly selected from outside haploblocks and genes. The estimated slopes for eight haploblocks fell into the 5% tail of the distribution for at least one of the ranges (Supplementary Fig. 11A). However, this approach did not examine temporal changes nor the combined signatures of selection over space and time. To do so, we ran generalized linear models comparing haplotype frequency with latitude, time (date of specimen in years), and range (North America vs. Europe; Fig. 5A; Supplementary Fig. 12; Supplementary Table 9; Supplementary Data 5–9). All but two (hb-chr6a and hb-chr7a) of the haploblocks showed significant changes over time, space, or both time and space. These patterns were robust to time being coded as discrete (historic vs. modern) or as continuous (by year; Supplementary Data 5). Most showed temporal changes either in their average frequency in one or both ranges, or in their relationship with latitude within each range, a pattern that is consistent with recent local selection on these haploblocks. Most of these haploblocks also showed significant associations with latitude in at least one range or timepoint, indicative of climate adaptation. For instance, hb-chr10, hb-chr14, hb-chr16b, and hb-chr17 all showed significant parallel latitudinal associations in both ranges in historic and modern samples. For hb-chr5b, haplotype frequency was negatively correlated with latitude in modern and historic samples from North America, as well as in modern European samples, consistent with climate-mediated selection. However, historic European populations did not display an association with latitude, which may reflect maladaptation during the initial stages of the European range expansion (Fig. 5A; Supplementary Data 7, 9).

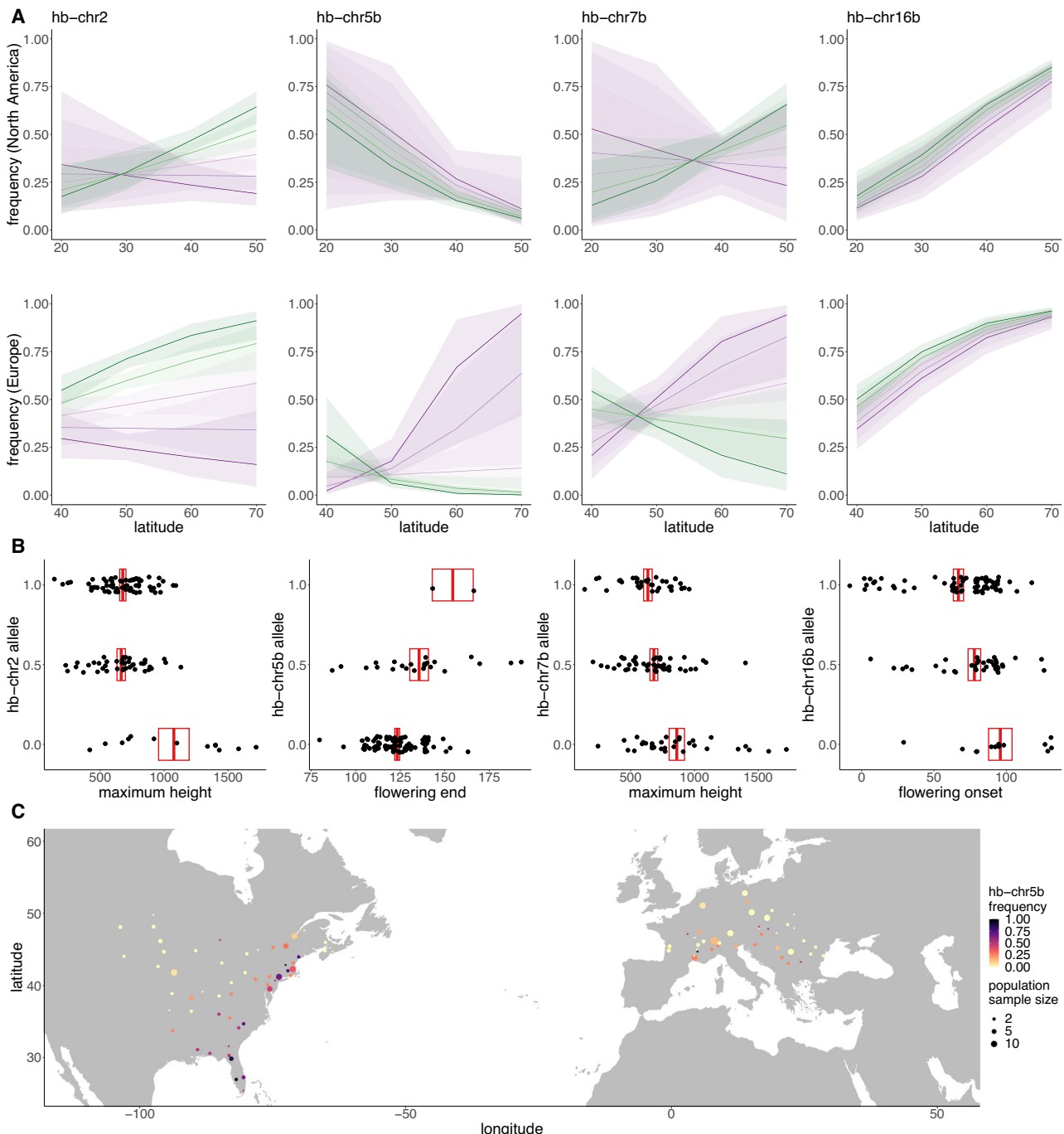

**Fig. 5 | Haploblock distributions and trait associations. A** Logistic regression models with error bands representing 95% CI of least-squares regressions (see Supplementary Table 9 and Supplementary Data 5–9 for model details) of haploblock frequency (allele 1) against latitude for four haploblocks across five time bins ranging from most historic (purple) to most modern (green). **B** Examples of significant associations between haploblock alleles and phenotypes ($n = 121$ biologically independent samples; boxes denote mean +/− SEM for each genotype). **C** hb-chr5b allele frequency in modern *A. artemisiifolia* populations. Source data are provided as a Source data file.

In one case (hb-chr7b) the significant latitudinal clines showed opposing yet significant slopes for each range in the modern samples, perhaps indicating contrasting patterns of local selection between the ranges. For hb-chr2 the latitudinal clines formed over time, with a substantial increase in frequency over time particularly at higher latitudes.

To further investigate whether European populations showed evidence of recent local selection on the haploblocks, we tested whether estimates of selection inferred from contemporary spatial data were associated with temporal changes in haploblock frequencies between historical and contemporary European populations. We used spatial variation in contemporary haploblock frequencies to estimate the relative strength of local selection on these haploblocks (see Supplementary Methods 1, 2). We specifically compared estimates of the maximum slope of latitudinal clines for each putative inversion's frequency to simple population-genetic models for clines at equilibrium between local selection and gene flow. In these models, cline slopes are proportional to $\sqrt{s}/\sigma$, where $s$ represents the strength of local selection for a given inversion and $\sigma$ is the average dispersal distance of individuals in the range[44] (Supplementary Methods 2;

Supplementary Data 7). While our estimates of selection are, therefore, scaled by the dispersal rate, dispersal should equally affect all inversions within a given range, allowing us to infer the relative strength of spatially varying selection for each putative inversion. We found that estimates of the relative strengths of local selection across the haploblocks in modern samples were correlated between the ranges, indicating parallel patterns of local selection along the latitudinal gradient ($r_{13} = 0.55$, $t = 2.41$, $p = 0.03$, 95% CI = {0.06, 0.83}). We also found that changes in the haploblock cline slopes between historic and modern time points within Europe were significantly correlated with our estimates of the relative strength of spatially varying selection of the haploblocks across the European range ($r_{13} = 0.86$, $t = 5.99$, $p = 4.55 \times 10^{-5}$, 95% CI = {0.61, 0.95}; Supplementary Fig. 11B). Such a pattern is consistent with a scenario in which historical European populations were not initially locally adapted (haploblock frequencies were initially far from local optima) and where the haploblocks subject to relatively strong local selection exhibited the greatest temporal changes in local frequency over the ensuing century. The same pattern was not observed in the North American native range, whose historic populations are likely to have been consistently closer to the local optima across the timescale of our analysis ($r_{13} = 0.25$, $t = 0.95$, $p = 0.36$, 95% CI = {−0.30, 0.68}). Cline slopes for latitude were also shallower in historic relative to modern European populations ($t_{10.38} = −3.09$, $p = 0.01$, 95% CI = {−0.16, −0.03}; mean absolute slope: historic Europe = 0.05; modern Europe = 0.14), but not so in North America ($t_{17.81} = −0.37$, $p = 0.71$, 95% CI = {−0.04, 0.03}), consistent with initial maladaptation in Europe, followed by adaptation to local climates.

The hb-chr2 haplotype frequency dramatically increased in many populations over time, particularly at mid to high latitudes, which encompasses the European populations and northern North American populations (Fig. 5; Supplementary Data 5, 9). Signatures of selective sweeps were observed in the hb-chr2 region in European populations in historic to modern comparisons of divergence and diversity (Fig. 3A; Supplementary Fig. 6). In this same region, negative outlier values of Fay & Wu's $H$ relative to the genome-wide pattern were identified in three of the five North American and six of the seven European locations, indicating an excess of high frequency derived SNPs and implicating recent positive selection (Supplementary Figs. 7, 8). To further distinguish the effects of drift from those of selection, we compared an empirical null distribution of allele frequency changes over time in Europe using 8303 SNPs matched for allele frequency (Fig. 3C). Only 11 (0.13%) of the 8303 loci in this null distribution showed an allele frequency change greater than hb-chr2. We then estimated strengths of selection for the putative inversion that are consistent with the observed increase in frequency within the invasive range (Supplementary Methods 1; Supplementary Table 10). With positive selection for the putative inversion, the estimated frequency shift at the center of the European range is consistent with a 2.4% difference in fitness (95% CI = {1.5%, 3.3%}; Supplementary Fig. 13) between individuals homozygous for the inversion relative to homozygotes for the alternative haplotype. Scenarios of balancing selection require stronger selection to explain the inversion frequency shift over time (see Supplementary Methods 1). This estimate is smaller than many empirical estimates of selection on individual loci in natural populations[45]. The greater timespan of our samples facilitated detection of strong signals of selection for loci that would otherwise be missed by the short time periods afforded by most temporal studies.

### Biological functions of haploblocks
Several analyses provide evidence for the biological function of these putatively adaptive haploblocks. From genome annotations, we found that haploblocks were collectively enriched for flowering-time pathway genes (Fisher's exact test $p = 0.002$), although individually only hb-chr5b was significantly enriched (Fisher's exact test $p = 0.001$; Supplementary Table 5), consistent with a large-effect flowering time QTL identified within this haploblock[42]. hb-chr2 was enriched for the

"recognition of pollen" gene ontology term, with 36 (17%) of the genome's 210 genes annotated with this term falling within this haploblock (Supplementary Data 4). hb-chr5a was enriched for genes with the "pectate lyase activity" term, including the top BLAST hit for *Amba1* (99.7% identity, *E*-value = 0), which encodes the *A. artemisiifolia* protein responsible for the majority of allergic reactions[46]. A detailed analysis of the hb-chr5a region reveals a cluster of six closely-related pectate lyase genes, which correspond with elevated XtX and XtX-EAA outlier windows in both ranges (Supplementary Fig. 14). hb-chr11 also overlaps the flowering time GWAS candidate *ELF3* (Fig. 2A, B), and the non-synonymous variant which displays strong patterns in GWAS is only observed on one of the haploblock genotype backgrounds. We also identified phenotypic associations with haploblocks by encoding haploblock genotypes into our GWAS pipeline. Significant associations ($p < 0.05$ Bonferroni-corrected for multiple testing across 15 haploblocks) were observed for four haploblocks, including for traits related to flowering time (Fig. 5B; Supplementary Table 11).

## Discussion
Here we present a new, chromosome-level phased diploid genome assembly for *A. artemisiifolia*. Using this high-quality reference and over 600 historic and modern *A. artemisiifolia* re-sequenced genomes from North America and Europe[11], we were able to uncover, at unprecedented temporal and spatial resolution, the evolutionary-genetic changes accompanying the recent and rapid invasion of Europe by this noxious pest. Previous work leveraged these re-sequenced genomes to understand population structure, invasion history as well as the microbial community of the species[11]. In this study, we have exposed dramatic genomic signals of parallel climate adaptation and discovered large structural variants that have played a significant adaptive role during this species' recent range expansion.

Our study system, while unique in many ways, yields results with important general implications for our understanding of the genetic basis of rapid adaptation to environmental change and the pervasiveness of parallel evolution in geographically widespread species. While invasive species are often envisaged to encounter novel selection pressures as they spread across alien landscapes, they must also readapt to similar environmental variation encountered in their native range, as haphazardly-introduced invaders are unlikely to be well-adapted to local conditions when the invasive populations initially expand across climatic gradients. Much of *Ambrosia artemisiifolia*'s European invasion lies within climatic extremes encountered across its native range (Supplementary Fig. 3). Despite this similarity in climatic variation, the patterns of parallel climate adaptation between native and invasive ranges are striking given the evolutionarily recent introduction of the species into Europe. As *A. artemisiifolia*'s invasion of Europe consisted of multiple introductions over a brief evolutionary time scale, these patterns are likely examples of collateral evolution[47], in which standing genetic variation in *A. artemisiifolia*'s native range has been co-opted for adaptation in and across the European invasive range. Parallel evolution is a hallmark of natural selection and parallel changes at the genetic level point to constraints and biases in the genetic pathways to adaptation that are evolutionarily achievable; when certain paths to adaptation are favored, such as when beneficial variants are already present in the population as standing variants, evolution will repeatedly draw on the same subset of genes to reach the same adaptive endpoints.

From herbarium specimens that were sampled throughout the course of *A. artemisiifolia*'s invasion of Europe, we observed an abrupt change in flowering and fruiting over time. Leveraging whole-genome sequences of herbarium samples across North America and Europe, we were also able to scan populations for temporal genomic signatures of selective sweeps. Although some populations have experienced shifts in ancestry over time in Europe[11], peaks against the genome-wide background provide compelling evidence for rapid local adaptation in

European populations, with the strongest genetic signals of rapid change over time corresponding to some of the strongest signatures of local adaptation in our spatial analyses, particularly windows in the region of the *ELF3* gene and hb-chr2. Further, these regions show parallel signals of climate adaptation in North America and are associated with adapting traits such as flowering onset. These multiple lines of evidence provide strong support that climate-mediated selection on phenology was pivotal in shaping the adaptive genetic landscape of *A. artemisiifolia* in Europe.

Large haploblocks (putative inversions) contribute substantially to these genetic signals of parallel adaptation. 27% of these haploblocks correspond to inversions segregating in our diploid assembly. We propose that these haploblocks maintain cassettes of co-selected genes that effectively segregate as single alleles of large effect[22,27], providing a genetic architecture suited to local adaptation in the face of high gene flow[11,19]. Consistent with this hypothesis, haploblocks are enriched for genes with particular biological functions, display associations with locally-adaptive traits, and carry signals of strong selection in both the native and invasive ranges. The evolution of inversions along environmental gradients has been reported in a range of species[23]. However, by investigating haploblocks in an invasive plant with extensive timestamped collections, we have demonstrated dramatic and adaptive evolutionary change of inversions under natural conditions, providing compelling evidence of strong and recent natural selection. These data have also allowed us to estimate selection for these variants, and we have shown that haploblocks with the strongest estimates of clinal selection are driven more rapidly toward their putative equilibria within the invasive range.

An important question during this era of environmental upheaval is the role of adaptation during range expansion and its necessity during colonization. Through our analysis of historic samples, we have shown that *A. artemisiifolia* was present in regions throughout Europe well before many of these adaptive variants became locally common, suggesting the species' extensive phenotypic plasticity may have facilitated its initial expansion. Strong local selection further improved the match between genotypes and local environments, even appearing to affect reproductive output in herbarium specimens. Many of the selected variants we identified are linked to traits that are key factors in the timing, length, and severity of the local pollen season (e.g., days to flowering onset, days to the end of pollen production, and biomass). Consequently, local adaptation has played a central role in shaping the allergy season in Europe and will likely continue to be critical as climate change and continued range expansion further amplify the damaging effects of this hazardous weed[48].

## Methods

### Genome assembly

Seeds collected from a wild *Ambrosia artemisiifolia* population in Novi Sad, Serbia (lat. 45.25472, lon. 19.91231) were sown in potting soil at a greenhouse facility at the Ringve Botanical Garden, NTNU University Museum (Trondheim. Norway). After 160 days of growth under stable light and watering conditions, young leaf tissue from mature individual plant "NSS02/B" was sampled and flash-frozen in liquid nitrogen. These tissues were then shipped to Dovetail Genomics for high molecular weight DNA extraction and library building.

DNA samples were quantified using Qubit 2.0 Fluorometer (Life Technologies, Carlsbad, CA, USA). The PacBio SMRTbell library (~20 kbp mean insert length) for PacBio Sequel was constructed using SMRTbell Express Template Prep Kit 2.0 (PacBio, Menlo Park, CA, USA) using the manufacturer recommended protocol. The library was bound to polymerase using the Sequel II Binding Kit 2.0 (PacBio) and loaded onto PacBio Sequel II. Sequencing was performed on PacBio Sequel II 8M SMRT cells generating 65.9 Gbp of data. These PacBio CCS reads were used as an input to Hifiasm[49].

For each Dovetail Omni-C library, chromatin was fixed in place with formaldehyde in the nucleus and then extracted. Fixed chromatin was digested with DNAse I, chromatin ends were repaired and ligated to a biotinylated bridge adapter followed by proximity ligation of adapter containing ends. After proximity ligation, crosslinks were reversed and the DNA purified. Purified DNA was treated to remove biotin that was not internal to ligated fragments. Sequencing libraries were generated using NEBNext Ultra enzymes and Illumina-compatible adapters. Biotin-containing fragments were isolated using streptavidin beads before PCR enrichment of each library. The library was sequenced on an Illumina HiSeqX platform to produce ~30X sequence coverage. The PacBio CCS reads and Omni-C reads were then used as input for Hifiasm to produce two haplotype-resolved assemblies (hap1 and hap2) using default parameters.

HiRise was used (see read-pair above) to scaffold each haplotype-resolved assembly. Each de novo assembly and Dovetail OmniC library reads were used as input data for HiRise, a software pipeline designed specifically for using proximity ligation data to scaffold genome assemblies[28]. Dovetail OmniC library sequences were aligned to the draft input assembly using bwa[50]. The separations of Dovetail OmniC read pairs mapped within draft scaffolds were analyzed by HiRise to produce a likelihood model for genomic distance between read pairs, and the model was used to identify and break putative misjoins, to score prospective joins, and make joins above a default threshold (Supplementary Fig. 1C). The NCBI[51] genome submission portal identified 30 (7.4 Mbp total) and 26 (6.4 Mbp total) scaffolds in haplotypes 1 and 2 respectively containing bacterial contamination which were subsequently removed from the final assembly.

We used GenomeScope v.2.0 to estimate the genome size and ploidy using 21mers identified in the reads with Jellyfish v.2.3.0[52]. Genomescope estimated the haploid genome size to be 1.04 Gbp using a diploid model (Supplementary Fig. 1A), a better model fit (95%) than the tetraploid model (91%), which also vastly underestimated the haploid genome size (497 Mbp). This finding was consistent with the smudgeplot produced by Genomescope, which also indicated diploidy (Supplementary Fig. 1B). The final assembly sizes for each haplotype were 1.11 and 1.07 Gbp, which were similar to the GenomeScope estimates. BUSCO (Benchmarking Universal Single-Copy Orthologs) v.5.1.3[31] analysis of each assembly using the eukaryota *odb10* dataset (Supplementary Table 1) demonstrated that both assemblies were complete with relatively low levels of duplication given the history of whole-genome duplication in the tribe.

To assess the presence of remnant haplotigs and other assembly artifacts, we mapped Illumina reads used in the reference genome assembly to haplotype 1 of the reference genome using Adapter-Removal v.2.3.1[53], BWA-MEM v.0.7.17[50], and Picard v.2.19.0 MarkDuplicates (https://broadinstitute.github.io/picard/), and measured average sequencing depth and heterozygosity of the alignment in non-overlapping 1 Mbp windows across the genome. Window depth was never greater than two times higher or 0.5 times lower than the mean, and furthermore regions of both low depth and low heterozygosity were distributed throughout the genome. The fact that there were no large regions with both low read-depth and low heterozygosity points to the success of the haplotype-resolved assembly (Supplementary Fig. 1D). Minimap2 was used to align each haplotype against itself and against one another, after filtering for alignments shorter than 10kbp and with fewer than 5000 matches, to identify homologous blocks that may represent haplotigs. This analysis revealed the presence of a misassembly where each assembly contained a region of scaffold 18 duplicated on scaffold 19, while the orthologous region was missing in the alternative assembly. This suggested that a section of scaffold 18 in each haplotype-resolved assembly had been incorrectly placed in the wrong haplotype (corresponding to chromosome 18 in haplotype 1 and chromosome 9 in haplotype 2). After making these manual corrections, the genetic map confirmed the continuity of these

chromosomes in each haplotype (Supplementary Fig. 15). The alignments of the final corrected assemblies within and between the assemblies further confirmed the continuity of the assemblies and the absence of haplotigs (Fig. 1B).

## Whole-genome resequencing samples

Whole-genome resequencing data used in this study have previously been described in Bieker et al.[11]. Modern samples were field-collected between 2007 and 2019, and historic samples were sequenced from herbarium specimens collected between 1830 and 1973. 121 modern samples with corresponding phenotype data collected by van Boheemen, Atwater, and Hodgins[14] were used for genome-wide association studies. 284 modern samples (from populations with a sample size >= 2) were used for environmental-allele associations. 97 historic and 100 modern samples divided into twelve populations were used for historic-modern population comparisons (Supplementary Table 8). For *ELF3* analysis, 26 samples from within 200 km of Berlin (15 historic and eleven modern) and 21 samples within 200 km of Quebec City (ten historic and eleven modern) were used. Genotyping and analysis of haploblocks was performed using 311 modern and 305 historic samples. For details of each sample see Supplementary Data 2.

## Sample alignment, variant calling, and filtering

FASTQ files from historic and modern *A. artemisiifolia* samples from North America and Europe[11] were aligned to haplotype 1 of our new reference genome using the Paleomix pipeline v.1.2.13.4[54], which incorporates AdapterRemoval v.2.3.1[53], BWA-MEM v.0.7.17[50], Picard v.2.19.0 MarkDuplicates (https://broadinstitute.github.io/picard/) and GATK v.3.7 IndelRealigner[55]. Mean depths of alignments ranged from 0.37X to 19.95X with a mean of 4.05X for historic samples, and 1.75X to 44.03X with a mean of 6.86X for modern samples (Supplementary Data 2). Variants were called in the higher-depth modern samples using GATK 3.7 UnifiedGenotyper[56] on all contigs greater than 100kbp in length. GATK VariantFiltration[55] and VcfTools[57] were used to filter variant calls. SNP and indel calls were separately filtered using GATK hard-filtering recommendations (SNPs: QD < 2.0, FS > 60.0, SOR > 3.0, MQ < 40.0, ReadPosRankSum < −8.0, MQRankSum < −12.5; indels: QD < 2.0, FS > 200.0, SOR > 10.0, ReadPosRankSum < −20.0, InbreedingCoeff < −0.8). In addition, SNPs and indels were separately filtered for sites with depth (DP) less than one standard deviation below the mean, and greater than 1.5 standard deviations above the mean. Individual genotypes were set to missing if their depth was less than three, then variants with greater than 20% missing across all samples were removed. Samples with greater than 50% missing variants were removed. For the remaining 311 modern samples, genotypes were phased and imputed using Beagle v.5.2[58].

## Genome annotation

To obtain RNA transcript sequences for annotation of the genome, after 160 days of growth additional samples of leaf, stem, flower, root, and branch were taken from individual "NSS02/B" and flash-frozen in liquid nitrogen. From these we extracted RNA from seven tissues (young leaf, old leaf, stem, branch, and three stages of development of the floral head) using a Spectrum Plant Total RNA Kit (Sigma, USA) with on-column DNA digestion following the manufacturer's protocol. RNA extracts from all five tissues were pooled into a single sample. mRNA was enriched using oligo (dT) beads, and the first-strand cDNA was synthesized using the Clontech SMARTer PCR cDNA Synthesis Kit, followed by first-strand synthesis with SMARTScribeTM Reverse Transcriptase. After cDNA amplification, a portion of the product was used directly as a non-size selected SMRTbell library. In parallel, the rest of amplification was first selected using either BluePippin or SageELF, and then used to construct a size-selected SMRTbell library after size fractionation. DNA damage and ends were then repaired, followed by hairpin adapter ligation. Finally, sequencing primers and

polymerase were annealed to SMRTbell templates, and IsoSeq isoform sequencing was performed by Novogene Europe (Cambridge, UK) using a PacBio Sequel II instrument, yielding 97,819,215 HiFi reads. To prepare the raw IsoSeq RNA data for downstream use in the annotation of the genome, we first identified the transcripts in the PacBio single-molecule sequencing data by following the IsoSeq v3 pipeline provided by PacificBiosciences (https://github.com/PacificBiosciences/IsoSeq). Briefly, the pipeline takes PacBio subread files as an input and undergoes steps of consensus generation, demultiplexing of primers, IsoSeq3 refinement, followed by a final clustering of the reads.

Prior to annotation of the genome, repetitive elements were identified using RepeatModeler v.2.0.1[59]. ProtExcluder v.1.2[60] was then run to remove any protein coding genes from the repeat library. RepeatMasker v.4.1.1[61] was used to mask the genome using the finalized repeat library (Supplementary Data 1). A large fraction of the genome consisted of repetitive sequence (66.51%; Fig. 1A). Retroelements were the largest class (39.13%), with long terminal repeats, particularly Gypsy (7.73 %) and Copia (18.82%), the most prevalent retroelements.

Genome annotation was performed using the MAKER v.3.01.03[34] pipeline. Genome assembly fasta file, the custom repeat library, IsoSeq clustered reads merged with a previously described transcriptome[11] (as expressed sequence tag [EST] evidence) and protein homology evidence from a plant protein database which combines the Swissprot plant protein database and NCBI Refseq for plants excluding transposable elements were used as the input files for the first run of the annotation pipeline. The custom repeat library was used to mask the repetitive regions. Additional regions with low complexity were soft masked using RepeatMasker v.4.1.1[61]. Gene predictors SNAP v.2013-11-29[62] and AUGUSTUS v.3.3.3[63] were trained by running iterative runs of MAKER as recommended by[34]. As the first round of annotation was based on the alignment of the EST evidence to the genome, *est2genome* option in the MAKER control file was set to allow MAKER to infer gene models directly from the EST evidence. After the completion of the first round of annotations, gene models with an AED (Annotation Edit Distance) score of 0.25 or greater and a length of 50 or more amino acids were retained and used to train SNAP v.2013-11-29[62] to obtain a SNAP hmm file. We then trained AUGUSTUS v.3.3.3[63] using BUSCO version 5.1.3[31]. First, training sequences were identified using the gene models predicted by MAKER from the first run by excising regions with mRNA annotations and 1000 bp on either side. These were used to run BUSCO using the embryophyte set of conserved genes. After training both SNAP and Augustus, MAKER was run again, with SNAP hmm and Augustus files. A total of three rounds of training for each gene predictor were run. We used the script genestats[64] to calculate the numbers and lengths of genes, exons, introns, and UTR (untranslated region) sequences present in the predicted gene models by the final MAKER run. We ran BUSCO v.5.1.3[31] with the eukaryota *odb10* lineage data set on the predicted transcript fasta file by MAKER to assess the quality and the completeness of the annotated genome.

For haplotype 1, a high-confidence gene set of 36,826 gene models with strong protein or transcript support was identified (Supplementary Table 3). Gene models were compared with *Arabidopsis thaliana* annotations (TAIR10 representative gene model proteins[65]) and the UniProtKB plants database using the *blastp* command in BLAST+[66]. Using an *E*-value threshold of $1 \times 10^{-6}$, 32,370 (87.9%) genes matched TAIR10 annotations and 28,092 (76.3%) matched UniProtKB. We identified 98.4% of the core eukaryotic genes amongst our annotated genes, 73.3% being single copy, 20% being duplicated and 5.1% fragmented compared to BUSCO markers present in the library *eukaryota_odb10.2020-09.10* (Fig. 1C). Gene ontology (GO) enrichment was assessed using GO terms from *A. thaliana* TAIR 10[65] BLAST results. To identify GO terms enriched among candidate lists, the R/topGO package[67] was used with Fisher's exact test, the 'weight01' algorithm, and a *p*-value <0.05 to assess significance.

In addition, annotations were cross-referenced with 306 *A. thaliana* FLOR-ID flowering time pathway genes[68]. 513 predicted *A. artemisiifolia* genes were matched to this dataset, representing 218 unique FLOR-ID genes. Enrichment of flowering time genes was also assessed in candidate gene lists using Fisher's exact test and a $p < 0.05$ threshold. The effects of imputed variants on predicted genes were estimated using SnpEff[69].

## Allele frequency outliers and environmental-allele associations

Imputed genotype data from modern samples were divided for between-range and within-range analyses in PLINK v.1.9[70], and a minor allele frequency threshold of 0.05 was applied within data subsets. For within-range analyses, sampling locations with fewer than two samples were excluded and allele frequencies were calculated for each sampling location, resulting in 1,150,328 SNPs across 143 samples and 43 populations in North America and 1,132,342 SNPs across 141 samples and 31 populations for Europe. Allele frequency outliers were identified within each range using the BayPass v.2.2 core model[37], with an $\Omega$ covariance matrix computed from 10,000 randomly-sampled SNPs that were located outside annotated genes and haploblocks, and pruned for linkage disequilibrium using a window size of 50 kb, a step size of 5 bp and an $r^2$ of 0.5 in PLINK[70]. To identify allele frequency variation associated with environmental variables within ranges, 19 bioclimatic variables were extracted for each sampling location from the WorldClim database[38] using the R/raster package[71]. Population allele frequencies were assessed for correlation with 19 bioclimatic variables using Kendall's $\tau$ statistic in R[72]. Genome-wide XtX and $\tau$ results were analyzed in non-overlapping 10 kbp windows using the weighted-Z analysis (WZA)[39], with the top 5% of windows designated outliers.

## Genome-wide association studies

Imputed genotypes from modern samples were filtered in PLINK v.1.9[70]. Non-SNP sites and sites with more than two alleles were removed. The 121 samples overlapping those phenotyped by van Boheemen, Atwater, and Hodgins[14] were retained (Supplementary Data 2), and sites with a minor allele frequency below 0.05 were removed, resulting in 1,142,278 SNPs for analysis. Genome-wide association studies (GWAS) were performed across 121 individuals from both North American ($n = 43$) and European ($n = 78$) ranges using EMMAX v.beta-7Mar2010[73], and incorporating an identity-by-state kinship matrix (generated in PLINK 1.9)[70] to account for genetic structure among samples. The kinship matrix was computed using 790,209 SNPs which remained after pruning for linkage disequilibrium using a window size of 50 kb, a step size of 5 bp, and an $r^2$ of 0.5. Candidate SNPs were identified using a conservative threshold of Bonferroni-corrected $p$-values <0.05.

## Phenotypic analysis of herbarium specimens

We conducted a trait-based analysis of herbarium specimens found in the Global Biodiversity Information Facility database (gbif.org 2021). We compiled information from all *A. artemisiifolia* European herbarium specimens for which there was a digitized image of the individual in the database alongside corresponding metadata (location and collection date). The collection date spanned 1849 to 2020 (median 1975) and comprised 985 specimens. We determined the stage of flowering (no male inflorescence present, only immature male inflorescence present, mature male inflorescence present) for each image. The presence of fruit was also recorded. The male inflorescence was used as an indicator of flowering as these structures are more visually prominent than female flowers and the onset of male and female flowering is highly correlated[14]. Male florets consist of prominent spike-like racemes of male capitula, and are found at the terminus of the stem, whereas female florets are observed to be in inconspicuous cyme-like clusters and are arranged

in groups at the axils of main and lateral stem leaves. The dates when the specimens were collected were converted to Julian day of the year. We conducted a generalized linear model with a binomial response and logit link (glm R). Both binary traits (presence of a mature male inflorescence; the presence of fruit) were included as response variables in two separate models. The significance of the effects were tested using the *anova* function (Car package R)[74] using type 3 tests. For both models, the predictors of latitude, day of the year, and collection year as well as all interactions were included. Non-significant interactions were removed in a stepwise fashion, starting with the highest order. Latitude of origin strongly correlates with flowering time in common garden experiments[14] and we expected northern populations to evolve early flowering relative to the start of the growing season to match the shorter growing seasons in these areas. As a result, if local phenology has evolved to better match the local growing season we predicted a collection year by latitude interaction, as the relationship between latitude and the probability of flowering in wild collected accessions should change over time when controlling for the day of collection.

## Historic-modern genomic comparisons

To identify targets of recent selection, we compared historic and modern samples from twelve locations (five locations from North America and seven from Europe; Supplementary Table 8). Historic samples were grouped based on age of sample and proximity to a modern population. Analyses were performed in ANGSD v.0.931[75] using genotype likelihoods. For each population location we calculated pairwise nucleotide diversity ($\theta_\pi$) for historic and modern populations separately, and $F_{ST}$ between historic and modern populations at each location. Statistics were calculated in non-overlapping 10 kbp windows, and windowed $\theta_\pi$ values were normalized by dividing by the number of sites in each window. At each location, windows with $\theta_\pi$ more than two standard deviations below the mean in both historic and modern populations were excluded from the analysis. We identified putative selective sweeps in each population as windows with extreme shifts over time in allele frequency as well as extreme reductions in diversity (i.e., windows in the top one percent of both $F_{ST}$ and $\theta_{\pi H}/\theta_{\pi M}$ distributions). To obtain further evidence for selective sweeps in these populations, we also performed genome scans of Fay and Wu's H in each modern population. We first generated an ancestral consensus sequence in ANGSD (-doFasta 2 -minMapQ 25 -minQ 20 -remove_bads 1 -uniqueOnly 1 -doCounts 1) from alignments of *Ambrosia carduacea* and *Ambrosia chamissonis* to our *A. artemisiifolia* reference genome. We then used this ancestral sequence in calculating Fay and Wu's *H* in 10 kbp windows in ANGSD.

## Temporal allele frequency shifts in candidate loci

In order to track allele frequency shifts over time, we estimated contemporary and historical allele frequencies of the *ELF3* nonsynonymous SNP and the haploblock hb-chr2, which are two candidate loci for recent selection in Europe. Both candidates showed evidence of local selection using spatial analysis of modern populations, as well as sweep signals in temporal comparisons of individual populations. These calculations were performed in geographic regions where this recent selection is believed to have occurred at both historic and contemporary timepoints. ANGSD[75] (-minMapQ 10 -minQ 5 -GL 2 -doMajorMinor 1 -doMaf 2 -doIBS 1 -doCounts 1 -doGlf 2) was used to calculate the allele frequency of the early flowering *ELF3* allele (11:41517231) in 15 historic and eleven modern samples from within 200 km of Berlin, whilst the frequency of hb-chr2 in Europe was ascertained using haploblock frequency estimates from across the European range (see below). To understand the magnitude of these allele frequency shifts relative to putatively neutral alleles elsewhere in the genome, we calculated a standardized measure of frequency change, $y_t$, using estimates of historic, $p_0$, and contemporary, $p_t$, allele

frequencies according to the following equation:

$$y_t = \frac{p_t - p_0}{\sqrt{t p_0 (1 - p_0)}} \qquad (1)$$

where $t$ is the number of generations separating the frequency estimates (equivalent to the number of years due to ragweed's annual lifecycle). As we show in Supplementary Methods 3, the distribution of $y_t$ estimates under neutrality are predictable and roughly independent of the initial frequency of each neutral variant once the loci with low-frequency initial minor allele frequencies are filtered out (Supplementary Fig. 16). To further assess if selection was the likely cause of temporal changes of the *ELF3* and hb-chr2 variants, we estimated the distribution of $y_t$ estimates computed from 10,000 randomly-sampled SNPs that were located outside annotated genes and haploblocks and pruned for linkage disequilibrium using a window size of 50 kb, a step size of 5 bp and an $r^2$ of 0.5 in PLINK v.1.9[70]. Prior to calculation of $y_t$, sampled SNPs were then filtered for a minor allele frequency >0.2 for hb-chr2 comparisons and MAF > 0.05 for *ELF3* comparisons (due to the low historic frequency of *ELF3* in historic Berlin populations), resulting in null distributions of between 6913 and 8303 SNPs. We then compared the distributions to the $y_t$ values of candidate adaptation loci to test whether candidate regions were more divergent than the putatively neutral distribution. As a point of comparison we repeated this analysis for hb-chr2 in North America and the *ELF3* allele in Quebec. As in Berlin, the *ELF3* allele is at high frequencies, but substantial temporal change was not expected as the populations were predicted to be closer to the equilibrium over the temporal sampling period in the native range. Samples within 200 km of Quebec City (ten historic and eleven modern) were pooled at both timepoints. Allele frequency changes of the 10,000 randomly-sampled SNPs and the non-synonymous *ELF3* allele were assessed as above.

## Haploblock identification

To identify signatures of large, segregating haploblocks across the genome, we performed local windowed principal component analysis with Lostruct v.0.0.0.9000[43]. Using SNP data from 311 modern samples, we extracted the first ten multidimensional scaling (MDS) coordinates across each chromosome in windows of 100 SNPs. These MDS coordinates were then plotted along each scaffold to observe regions of local structure, indicative of segregating haploblocks. We focused on outlier MDS signals that overlapped parallel outlier windows for both XtX and at least one environmental variable, and also showed well-defined boundaries indicative of chromosomal inversions. We tested for additional evidence of inversions using PCA of MDS outlier regions and kmeans clustering in R[72] to identify regions containing three distinct clusters representing heterozygotes and two homozygotes. In addition, we assessed heterozygosity from genotype data in each haploblock region and in each modern sample, and measured linkage disequilibrium (the second highest $r^2$ value in 0.5 Mbp windows) across each scaffold bearing a haploblock for all modern samples and for modern samples homozygous for the more common haploblock genotype using scripts from Todesco et al.[26].

## Haploblock frequency changes over time and space

For fifteen candidate inversions, a local PCA of each region and kmeans clustering was then repeated in PCAngsd[76], so as to allow genotype estimation of these haploblocks in 305 historic samples alongside the 311 modern samples. For this local PCA we used only the chromosomal regions already defined as haploblocks in order to obtain population wide clustering for both historic and modern datasets, which we then used to infer haploblock genotypes. We also conducted a PCA on 10,000 SNPs randomly-sampled from the 311 modern genomes that were located outside annotated genes and haploblocks, and pruned for linkage disequilibrium using a window size of 50 kb, a step size of 5 bp and an $r^2$ of 0.5 in PLINK[70]. Following this, we used generalized linear models (glm R) to assess how haplotype frequency (binomial response) changed over time and space. A count of each haplotype at a geographic location and year was the binomial response variable and time period (historic or modern), range (North America or Europe), latitude, and all interactions between these three main effects were used as predictors. Non-significant interactions were removed in a stepwise fashion, starting with the highest order. PC1 from the PCA of 10,000 randomly-sampled SNPs was included as a covariate to control for the effects of population structure on haplotype frequency. We tested the significance of the effects in our model using the *anova* function (Car package R)[74] with type 3 tests. Significant differences among groups for means or slopes were tested with the emmeans package using an FDR correction[77]. To determine if the classification of samples into modern or historic timepoints influenced our results we ran a second set of generalized linear models examining haplotype frequency as a function of collection year, range (North America or Europe), latitude, and all interactions between these three main effects as well as PC1, using the same approach as above. For interactions involving two continuous variables (i.e., latitude and year) we tested if the slope estimates of one variable were significant at specific values of the other using the package emmeans. This allowed us to estimate when and where the haplotype frequencies were changing. The results from both approaches (time as two categories or time as continuous) provided qualitatively similar patterns.

We estimated the relative strength of selection on haploblocks along the latitudinal clines in modern North American and European populations using slopes from logistic regressions (see Supplementary Methods 2). Specifically, we used generalized linear models to estimate the slopes of the regression for each range and time point (modern or historic) combination (group). A count of each haplotype at a geographic location and collection year was the binomial response variable and time period (historic or modern), range (North America or Europe), latitude, and all interactions between these three main effects were used as predictors. All interactions were retained in the model and slopes and their confidence intervals estimated for each group using the function emtrends (emmeans package R[77]; Supplementary Table 9). PC1 was included as a covariate to control for the effects of population structure on haplotype frequency. We expected the slopes to be shallower in the historic versus the modern European group, but similar across timepoints in North America. To test this, we used a t-test and compared slopes for modern and historic timepoints in each range. We also expected that the magnitude of change in the slope over time would be the greatest in haploblocks showing the largest estimates of selection in Europe (Supplementary Data 7). We estimated the relative strength of selection for the modern European range for each haploblock and tested if the absolute change in slope for each haploblock was correlated with this estimate. We also examined if there was a correlation in the relative strength of selection for modern North American and European haploblocks, which would indicate parallel selection along the cline in each range.

We compared our slope estimates of the haploblocks to the genome-wide distribution in each range using 10,000 randomly selected SNPs outside of genes and haploblocks. We did this to determine if our haploblocks showed stronger latitudinal patterns than the majority of SNPs, in one or both ranges, which may be indicative of spatially varying selection. For the modern samples in each range (North America or Europe), we fit a generalized linear model with latitude as the only predictor. We did this for each null SNP and each haploblock that was statistically associated with latitude.

## Recombination rates in haploblocks

The haploblocks show multiple genomic signatures of reduced recombination. To confirm this we analyzed recombination rates in genetic maps. If the haploblocks were caused by global reductions

in recombination rate (e.g., the region was found in an area with generally low recombination such as a centromere), all maps should show reduced recombination rates. However, if inversions were the cause, recombination would only be suppressed in genotypes heterozygous for the inversion, while homozygous individuals would not show suppressed recombination. To determine if there were genotype-specific reductions in recombination rate in the haploblocks, which would be consistent with inversions, we made use of three previously generated genetic maps[42]. Markers were generated using genotype by sequencing and alignments to the haplotype 1 of our diploid reference genome. Details of the sequencing, alignments, and variant calling can be found in Prapas et al.[42]. We developed sex-specific genetic maps (i.e., maps for the maternal and paternal parent) using Lep-MAP3[78] for each chromosome of interest and in each mapping population (an F1 mapping population and two F2 mapping populations). Multiple maps were constructed since the haploblocks may have been segregating in different frequencies in the parents of the mapping populations derived from outcrossing. For the recombination rates, linkage map construction was constrained by the physical order of the markers along each scaffold of interest. Genetic distance (cM) was plotted against physical position along the chromosome for each map and the intervals of the QTL and the boundaries of the haploblocks were visualized and inspected for reduced recombination compared to the rest of the scaffold. We also used the genetic map (pink family) to confirm the OmniC scaffolding, and assess the accuracy of the manual correction of chromosome 18 in both assemblies (Supplementary Fig. 15).

### Reporting summary

Further information on research design is available in the Nature Portfolio Reporting Summary linked to this article.

## Data availability

Sequences used in reference genome assembly and annotation are available from NCBI under BioProject ID PRJNA819156. The phased diploid genome assembly is available from NCBI under BioProject IDs PRJNA929657 and PRJNA929658. The haplotype 1 gene annotation GFF file is available from Figshare [https://doi.org/10.6084/m9.figshare.19672710.v1]. Individual sample resequencing data are available from ENA under BioProject IDs PRJEB48563, PRJNA339123 and PRJEB34825. Source data are provided with this paper.

## Code availability

All code for analyses performed in this work is available from Github [https://github.com/pbattlay/ragweed-selection].

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

## Acknowledgements

We thank Greg Owens & Michael Whitlock for discussions, and Samuel Yeaman and Sarah Otto for feedback on the manuscript. We are grateful to François Bretagnolle, Myriam Gaudeul, Heinz Mueller-Schaerer, Gerhard Karrer, and Bruno Chauvel for their assistance in obtaining many of the samples upon which this study is based, and Marie Brunier, Fátima Sánchez Barreiro, Yohann Roy, Luna Forcioli, Jacqueline Y. Lee for assistance during lab work. We thank Vibekke Vange and the NTNU Ringve Botanical Garden team for their assistance in cultivating the plants. Some sequencing services were provided by the Norwegian Sequencing Centre, a national technology platform hosted by the University of Oslo. Some sequencing was performed by the NTNU Genomics Core Facility. Genome scaffolding was performed by Dovetail Genomics. Some analyses were performed on resources provided by Sigma2, MASSIVE M3, and ComputeCanada high performance computing platforms. We kindly thank the curators from the following herbaria for allowing us to destructively sample their precious collections: B, BR, BRNU, C, FI, G, GH, GOET, GZU, HBG, I, IASI, JE, L, LD, LY, MARS, MASS, MO, MPU, NEBC, NEU, NY, P, PH, PR, PRA, PRC, QFA, S, STU, TRH, UPS, US, W, WU. This research received support from an NTNU Onsager Fellowship award, Norwegian. Research Council Young Research Talents award 287327, and a SYNTHESYS Project award (www.synthesys.info, financed by European Community Research Infrastructure Action under the FP7 "Capacities" Program) to M.D.M., a FORMAS (2016-00453) & Carl Trygger Foundation for Scientific Research (grant CTS 14.425) to R.S., and an ARC DP220102362 and DP180102531 and HFSP RGP0001/2019 to K.A.H.

## Author contributions

P.B.: Software, formal analysis, investigation, data curation, writing—original draft, writing—review and editing, visualization. J.W.: Software, formal analysis, investigation, writing—review and editing, visualization. V.C.B.: Software, investigation, data curation. C.L.: Investigation, resources. D.P.: Software, formal analysis. B.P.: Software. S.C.: Investigation. L.v.B.: Investigation, resources. R.S.: Resources. N.P.d.S.: Software, visualization. A.S.: Investigation, resources. B.K.: Investigation, resources. K.A.N.: Investigation, resources. L.R.: Writing—review and editing. T.C.: Methodology, formal analysis, investigation, writing—original draft, writing—review and editing. M.D.M.: Conceptualization, methodology, resources, writing—review and editing, supervision, project administration, funding acquisition. K.A.H.: Conceptualization, methodology, software, formal analysis, investigation, resources, writing—original draft, writing—review and editing, supervision, project administration, funding acquisition, visualization.

## Competing interests

The authors declare no competing interests.
