## [Peer Review File · Nature Communications]

Large haploblocks underlie rapid adaptation in the invasive weed *Ambrosia artemisiifolia*Reviewers' Comments:

Reviewer #1:

Remarks to the Author:

Invasive species is an interesting system for adaptation study. The authors report a new chromosome-level reference genome assembly of the annual weed *Ambrosia artemisiifolia* and over 600 sequenced genomes (including herbarium samples), and investigate selections from spatial and temporal aspects. In general, this manuscript is meaningful for evolutionary biology and weed science. I do have several criticisms.

Why did not the authors present genome assembly as a section in RESULTS? It is definitely very important in this MS. The accuracy of assembly directly affects the subsequent analyses. I did not see too much information for 600 resequenced genomes either in the main text.

In the Methods, the authors claimed that the HiFi assembly yielded a 2.1-Gb genome, while the estimated genome size is ~1.2 Gb. I understand that heterozygosity results in larger assembly. But I don't quite understand 2.1 vs 1.2 (nearly twice). I strongly suggest that the authors perform genome size estimation for the sequenced accession by flow cytometry or K-mer. *Ambrosia artemisiifolia* is polyploid? The authors also need to confirm the ploidy of the accession you de novo sequenced.

More evidences should be provided for chromosomal inversions of the seven large haploblocks.

Reviewer #2:

Remarks to the Author:

In this manuscript, Battlay et al. assembled the genome of *Ambrosia artemisiifolia*, an invasive weed native to North America. The authors performed temporal and spatial genomic scan analyses with modern and historic samples from the native and recently-invaded Europe ranges. They identified genetic changes, especially large haploblocks, associated with the rapid adaptation to invasive environments.

Overall, I think this study system is very comprehensive, and the findings are essential for us to understand the rapid adaptation of invasive species. However, I suggest more analyses or discussions make the logic of this work more congruent.

My major concern is as follows.

As described in the manuscript, standing genetic variation in native range has been co-opted for adaptation in and across the invasive range. The authors argue that this adaptation is parallel. However, using parallel evolution to explain such results may not be accurate. It could be collateral genetic evolution. One review written by David L. Stern is relevant (*Nature Reviews Genetics*, v14, p751–764, 2013).

This study mentioned that the herbarium samples could be traced back to 1830. Do these herbarium samples over a large time span? This manuscript does not describe the collection time of herbarium samples but uniformly classify them as historic. It should be clarified if all historic samples have the same collection time (the same year or a similar year). Otherwise, the authors should be cautious about biases in allele frequencies of each historical population. In addition, analysis results (e.g., historic vs. modern F_{ST} and θ_{PIH}) related to allele frequencies in historic locations may be affected. The authors performed ABBA-BABA analysis using the same genus species and found that the source of variation in some haploblocks was introgression. But this result is no longer discussed. Adaptive introgression can be a driver of local adaptation. Then a question to the author is, could this affect their interpretation of their results?

Line 435-437: It was confused that which version of BUSCO was employed in this study, as you mean 253 complete genes is 99.2% of the total? Besides, I think the percentage of the number of duplicated genes (45.1%) was suspicious for a high-quality genome assembly in general. Please check and describe detailly.

Line 418-426: As you mentioned that the hifiasm assembled heterozygous contigs in one set of genome sequence, and the Purge_dups was always used to remove duplicated regions. However, the validation only based on the reduction of duplicated BUSCO genes was not convincing. It is necessary to do more effort on the validation on the genome assembly as it influenced the accuracy of the subsequent analysis directly. Firstly, self-to-self contigs alignment should be made to check the redundancy of genome assembly in a dot-plot. It is also essential to check the sequence depth and the heterozygosity using Illumina reads as the heterozygous contigs tend to show higher sequence depth and heterozygosity. Tool like GAAP (<https://github.com/zy-optimistic/GAAP>) could be employed to verify the redundancy. Command as follow: "gaap snvcov -r genome.fasta -i ngs1.fastq -I ngs2.fastq -l tgs.fastq -t threads".

Additional minor comments:

1. Line 36-37: "We show highly significant levels of parallelism between Europe and North America, with upwards of 11.7% of climate adaptation signals shared between ranges." This sentence is not clear. I couldn't find the result corresponding to "11.7% of climate adaptation signals" in the text. Please consider re-writing it.
2. Line 111: "alignments of over 600 modern and historic genome sequences". I found the usage of these samples in the manuscript to be unclear. From the Method section (Line 453), I know that the number of modern samples is 311. However, in the Result section, the authors performed genome scans for population allele frequencies among 284 modern samples (Line 127-129) and performed GWAS on 121 samples (Line 137). The usage of historic samples is even more ambiguous. Please describe them clearly.
3. Line 137: The author should give the full name of GWAS.
4. Line 142-143: "Candidate SNPs in ELF3 are restricted to high-latitude populations in both ranges." I noticed that the latitude ranges for North American and European are not equivalent. Due to the climatic characteristics of the two continents, should the authors discuss the temperature difference between the two ranges at the same latitude? In addition, it would be better to give references for ELF3.
5. Line 168: Does the legend at the bottom of Figure 1B refer to the number of samples in the population? The authors should change the breaks to integers.
6. Line 207: "... populations within a 200km radius of Berlin...". Please describe the number of these populations and list the ELF3 frequencies. It would be better to mark "200km radius of Berlin" in Figure 1B.
7. Line 223: The x-axis of Figure 2B lacks ticks and tick labels. It would be better to clearly show the size and position of the HB448 on the scaffold448.
8. Line 240-241: Please list the name and size of the seven haploblocks.
9. Line 245-249: I found the results of introgression and recombination rates unclear. Please replace "some" with the exact number (Line 246). "Six haploblocks showed evidence of...". It would be better to enumerate the six haploblocks or the one with the exception.
12. Line 406: please add the citation of hifiasm.
13. Line 670-678: There seems puzzling to choose such a large window size (10 Mb). The authors should explain the reason.

Reviewer #3:

Remarks to the Author:

Overview

The authors assembled a draft genome of *Ambrosia trifida* from a field population. They re-sequenced collections from north America and Europe for a GWAS in an attempt to see genomics regions under selection in both the native and invaded range. They identified one such locus that housed a gene that encodes for flowering time (ELF3) whose allele frequency changing with latitude. They were also able to identify several other loci under selection that were unique and/or shared between the two groups. Using historic and modern collections, they also identified a selective sweep possible large inversion in one of their scaffolds indicating a haploblock influencing local population structure. Globally, they were

able to identify several other haploblocks whose alleles frequencies did (or did not) correlate with latitude. Some of these blocks varied the same in historic collections while others diverged.

General Comments

Overall I think the writing is excellent, extremely detailed and the work impressive. However, I think the story is inflated and the message is muddled by jargon and lots of intermingled data references. I wish the introduction more clearly stated what it is they are trying to test and whether it was sufficiently answered. I think the story is interesting and relevant but lacks clarity. Furthermore, I think a lot is missing in the methods about what samples were taken where and when. It is hard to know exactly how to interpret the data without a clear idea of how many samples were being taken from each location (NA vs Europe) and modern versus herbarium. Again, I think the authors did an amazing job with data analysis and collection but did not translate exactly what it all means, what was exciting about the work, and how they came to those conclusions. One last note, I think it is critical that your assembly and associated annotation files be made available upon publication. Many authors are merely depositing raw reads on NCBI to obfuscate their work. Please make sure to clearly state what data is going to be available to others.

Line Comments

Lines 58-62: There is a lot of absolutes in here. For instance, while it is often the case that an invasive species has a broad native range or that it stress resilient, it is not always. Some species have narrow ranges until they escape predation in a new environment, sometimes they are able to invade new ranges precisely because they have "comfortable" conditions and are not particularly resilient. I caution against saying things like "Invasive species also inhabit broad and climatically diverse ranges" because you are saying this has to be true in order for an invasion to be successful, which it is not.

Lines 83-84: "High connectivity"? Do you mean high gene flow or that their patches are contiguous? This term is not clear to me.

Ln124-135: But how different is Europe from North America in terms of latitude, precipitation, temperature, etc? I would not consider one more extreme than the other and imagine that much of the climate is the same in both the native and invaded region. It seems that you are constantly alluding to an idea that the invaded range is "Extreme" but it is not. For instance, If a plant is well adapted to invading disturbed soybean fields in the us, the same is going to be true in Europe, Asian, and Africa.

Figure 1C: Why are the scales on the y axis for North America and Europe drastically different?

Ln 182: A shift in what direction? Earlier in the year?

Ln 185-186: "more recently collected plants were more likely to be flowering than older historic specimens." Were these collected at approximately the same time of year then? its not exactly clear.

Ln 202: What does this rapid adaptation depend on? I imagine for rapid adaptation to occur, the introduced plants must have sufficient diversity for adaptation to occur. What is genetic diversity like in the invaded and native range for this species (I believe it is primarily out crossing?) What exactly is it maladapted to? What in the invaded range is selecting these plants so hard since I imagine precipitation, light, and temperature in continental Europe to be similar to southern Canada and northern US.

Ln 351: Why would it be striking? Assuming that the same genetics were in both the invaded and native range and the climate in Mainland Europe and North America are relatively similar, would you not expect similar patterns and convergent evolution?

Ln 362: Couldn't inversions be verified in your various plants by looking at read-pair breaking in your resequencing data? It is relatively easy to confirm inversions using low-coverage resequencing with either PacBio or Paired end Illumina.

Ln 381: You use the word plasticity but I am not sure if you are referring to phenotypic plasticity, genotypic plasticity, genetic diversity or some combination of these.

Ln 437: I would like to see a better summary of the genome assembly and many more statistics to help evaluate its completeness. Though I trust you I would like to know more about the predicted genome size, number of chromosomes and how close you are to achieving this. It is unclear from the limited information in the supplementary tables.

Ln391: I am shocked there isn't an inclusion of which samples were collected from where and when. Maybe this is in the supplemental but understanding the time and shape of the populations collected will help with the overall evaluation of evolution. This is especially true if samples from the herbarium predate the use of herbicides to control this plant as I can imagine modern agriculture would apply many novel stresses that supersede climate. Furthermore, it is unclear from the methods if the samples were taken from open space, agriculture, and/or some other areas. Again, this may influence the downstream analysis but it is unclear.

Reviewer #4:

Remarks to the Author:

This study investigates local adaptation in the invasive weed *Ambrosia artemisiifolia*. A new, good quality reference genome was assembled for this species of worldwide importance. An impressive set of over 600 samples were sequenced. This material includes contemporary populations as well as historical material from herbarium and covers both the native range (North America) and the main invasive range (Europe). Various methods including phenotype scoring from herbarium images, genome-wide association mapping and different genome scan analyses were implemented to detect genomic regions associated with local adaptation.

Analyses were very thoroughly conducted and revealed parallel changes in the native and invasive ranges as well as congruence between the spatial and temporal patterns of genetic changes at a few genomic regions, including some large haploblocks. This convincingly demonstrates that a few genomic regions with large effects have contributed to local adaptation. Further, the temporal sampling scheme demonstrated that local adaptation was ancient in the native range, whereas in Europe, initial maladaptation was followed by rapid adaptation to local climates.

I have only few concerns about this study. My main concern is about the comparison of historic and contemporary populations.

First, the plant material is not described in details. Readers are referred to reference #10, which is not satisfactory, as this is not a published paper but a bioRxiv deposit. More details are needed about the sampling locations, dates of sampling and number of individuals per population (for both herbarium and contemporary populations). This could be provided as a supplementary material. The comparison of historic vs modern population was based on F_{st} , on the ratio of historic to modern nucleotide diversity, and on the modelling of allele frequency shift in time, for pairs of historic and modern populations said to have been sampled at the same locations. This approach seems to rely on the hypothesis that the modern populations that were sampled descended from the historic populations retrieved from herbarium, and that rapid adaptation has occurred in the time interval between the two samplings. This scenario implies that no extensive migration nor population

replacement have taken place. This seems a precarious hypothesis for an invasive plant that thrives in perturbed environments. The authors should give more details regarding their criteria for selecting the 16 pairs of populations considered. They possibly had some historical records in favour of the stability of populations over time? Also, what are the time intervals between historical and modern populations? Are these intervals similar for all the considered pairs?

In the other hand, it may well be that historic populations were composed of haphazardly introduced, maladapted genotypes that were subsequently replaced by more adapted genotypes, so that there is no ancestry relationship between paired historical and modern samplings. I think that this scenario will also leave a signature consistent with a selective sweep, and probably a signature of a strong selective sweep in the case of population extinction followed by replacement. However, in my mind, this process of demographic turn-over and sorting is different from rapid adaptation via selective sweep, and should be distinguished as such.

Other comments

I.151. I don't quite understand the rationale behind combining consecutive windows to account for linkage disequilibrium (more a personal thought, this does not necessarily necessitate any change)

I. 245-247. The authors implemented ABBA-BABA tests using either *Ambrosia trifida* or *Ambrosia psilostachya* as putative donor species and found some evidence for introgression. This result is presently very shortly and not discussed. It is not at all essential for the study and could be omitted. On the other hand, it raises some questions: to my knowledge, hybridization between *A. artemisiifolia* and either *A. trifida* or *A. psilostachya* has not been described in the literature. *A. psilostachya* is reproducing almost exclusively vegetatively, and, if I'm correct, its distribution does not overlap with that of *A. artemisiifolia*. Hybridization between these species seems rather unlikely and I wonder whether this could be incomplete lineage sorting rather than introgression.

L. 248 From Figure S10, it seems that evidence for a reduced recombination rates across haploblocks varies much depending on the genetic map considered. Hence, it is not clear to me, how the authors could conclude that six of the haploblocks showed reduced recombination consistent with inversions.

L. 252-253 "we genotyped all historic and modern samples by local, divergent population structure in haploblock regions" I find this part of the sentence quite unclear, and maybe the authors should rephrase in a more explanatory way. From the material and methods, I understood that the genotype of each sample was assigned from k-means clustering into three groups (the two homozygotes and the heterozygotes), using the first two components of a PCA. From fig S7 it seems that separation into three groups is not always clear-cut. This suggests that there was quite a lot of remnant genetic variation with each collapsed "genotype". I find it hard to assess whether ignoring this underlying genetic variation may have biased the tests of statistical association with latitude or time in one way or the other. Do the authors have any view about this?

I. 314 and I. 331: This is rather speculative but I wondered whether selection for enhanced dichogamy, or selection for traits that favour allogamy could have taken place during invasion of Europe? This could have favoured admixture, generating genetic variation useful for local adaptation.

Material and Methods

L. 439 and following: Sample alignment, variant calling

The genome re-sequencing method, obtained read coverage and approach for variant calling (seems to be a probabilistic approach using *angds*) are not explicitly stated.

REVIEWER COMMENTS

Reviewer #1 (Remarks to the Author):

Invasive species is an interesting system for adaptation study. The authors report a new chromosome-level reference genome assembly of the annual weed *Ambrosia artemisiifolia* and over 600 sequenced genomes (including herbarium samples), and investigate selections from spatial and temporal aspects. In general, this manuscript is meaningful for evolutionary biology and weed science. I do have several criticisms.

Why did not the authors present genome assembly as a section in RESULTS? It is definitely very important in this MS. The accuracy of assembly directly affects the subsequent analyses.

We thank the reviewer for pointing out this oversight. We have now added a section to the results describing the reference genome assembly (lines 125-140), which we feel improves the clarity and readability of the work

I did not see too much information for 600 resequenced genomes either in the main text.

We again thank the reviewer for identifying this oversight, and have now included a description of the resequenced samples in the Materials and Methods (lines 519-530) supported by table S4.

In the Methods, the authors claimed that the HiFi assembly yielded a 2.1-Gb genome, while the estimated genome size is ~1.2 Gb. I understand that heterozygosity results in larger assembly. But I don't quite understand 2.1 vs 1.2 (nearly twice). I strongly suggest that the authors perform genome size estimation for the sequenced accession by flow cytometry or K-mer. *Ambrosia artemisiifolia* is polyploid? The authors also need to confirm the ploidy of the accession you de novo sequenced.

We understand the concerns of the reviewer and had the same thoughts when we first examined the initial genome assembly, although this species is known to be diploid (e.g., Bassett & Crompton [1975] *Canadian Journal of Plant Science* 55[2]), and we have never identified polyploid populations despite our extensive molecular work on this species. We used the software GenomeScope 2.0 and an analysis of 21-mers identified in the sequence reads used for the assembly to estimate the size of the haploid genome, as well as the ploidy of the sequenced genome (fig. S16). The smudgeplot of heterozygous k-mer pairs clearly supports diploidy (fig. S17). Further, the estimated haploid genome size, using k-mer coverage distributions when diploidy is assumed, is 1.04Gbp. However, this genome is also estimated to be highly heterozygous, and this likely accounts for the near doubling in size of the initial assembly. We have now included this information in the manuscript (lines 478-484). In support of this analysis, when Hifiasm is run in diploid mode using both the HiFi reads and OmniC reads for phasing, it produces two phased haplotypes, each of which is approximately 1 Gbp in length, largely consistent with the size of our haploid assembly. We are working towards a phased diploid assembly for a future paper, but these ongoing analyses have not been included in this manuscript. Finally, our final genome size is similar to a previous, but significantly more

fragmented draft of the ragweed genome, based on sequencing a different individual from a different population, described in Bieker *et al.* (in press) *Science Advances*. Therefore we are confident that the individual used was not polyploid and that the assembly accurately reflects the expected haploid genome size.

More evidences should be provided for chromosomal inversions of the seven large haploblocks.

In response to this suggestion, we conducted a further analysis that again supports the hypothesis that the haploblocks are inversions. Specifically, we have now added to the manuscript an analysis which demonstrates regions of increased linkage disequilibrium corresponding to haploblock regions in our 311 modern samples (fig. S9 and lines 266-267; 727-731). As a result we believe that we provide substantial evidence that the haploblocks represent inversions, however previously this evidence was mostly presented and discussed in the Methods section, which may not have been the most appropriate place for them. We have now included a summary of the evidence that the haploblocks are inversions in the Results (lines 262-274). Specifically, we have demonstrated the following:

- 1) PCAs of haploblocks regions segregate into three clusters (consistent with the two homozygous genotypes and the heterozygous genotype).
- 2) The putative heterozygous genotypes have higher heterozygosity than the homozygote genotypes.
- 3) The haploblocks demonstrate higher LD within the haploblock region, and this difference tends to be reduced in the common homozygous genotype (new analysis), which is the expected pattern if the haploblock was caused by an inversion.
- 4) Multiple genetic maps demonstrate reduced recombination across haploblock regions. However, some maps also show recombination within haploblock regions. This map-specific reduction in recombination would occur if a structural variant, such as an inversion, were suppressing recombination between heterozygotes in some maps, but homozygosity were allowing recombination in other maps. If these haploblocks were caused by global reductions in recombination in these regions, we would not see this pattern, as all maps would show reduced recombination in the haploblock regions.

Several papers have identified these same signatures of inversions in other taxa and then confirmed their inversion status using other approaches (e.g., linkage maps or HiC data) further supporting our hypothesis (Li & Ralph [2019] *Genetics* 211[1]); Todesco *et al.* [2020] *Nature*; 584[7822]). Consequently, other published articles are now identifying these genomic signatures and reporting them as inversions without the presentation of evidence based on linkage maps or HiC (e.g., Mérot *et al.* [2021] *Molecular Biology and Evolution* 38[9]; Whiting *et al.* [2021] *PLoS genetics* 17[5]; Cao *et al.* [2022] *Communications Biology* 5[1]). We also note that we have been careful to describe the regions as 'haploblocks' or 'putative inversions' throughout the manuscript.

Presently, such sequence-based evidence of the inversions using available data is not possible. Such data (e.g., long reads) are only currently available for the individual used to construct the reference, and the reference individual would need to be segregating for the haploblock to be

verified. Based on heterozygosity within the reference (fig. S19), there are some haploblocks that may be heterozygous (HB448). However, when we investigated those regions in our reference, the breakpoints fell in repeat regions. Therefore our attempts to map or assemble the reads accurately across the likely breakpoints failed, even when we used the long, high-fidelity PacBio reads. We hope to sequence alternative homozygote genotypes in the future using HiC, or to map their marker order using optical mapping, as these data would confirm the presence and type of structural variant in these regions, but this would require substantially more data. Moreover, the absence of these data does not detract from the novelty of our overall findings demonstrating substantial and recent climate-mediated selection on these haplotype blocks over time and space.

Reviewer #2 (Remarks to the Author):

In this manuscript, Battlay et al. assembled the genome of *Ambrosia artemisiifolia*, an invasive weed native to North America. The authors performed temporal and spatial genomic scan analyses with modern and historic samples from the native and recently-invaded Europe ranges. They identified genetic changes, especially large haploblocks, associated with the rapid adaptation to invasive environments.

Overall, I think this study system is very comprehensive, and the findings are essential for us to understand the rapid adaptation of invasive species. However, I suggest more analyses or discussions make the logic of this work more congruent.

My major concern is as follows.

As described in the manuscript, standing genetic variation in native range has been co-opted for adaptation in and across the invasive range. The authors argue that this adaptation is parallel. However, using parallel evolution to explain such results may not be accurate. It could be collateral genetic evolution. One review written by David L. Stern is relevant (*Nature Reviews Genetics*, v14, p751–764, 2013).

We agree that the term ‘parallel evolution’ is imprecise and can mean different things to different readers, although we note that it is common to use it to describe repeated adaptation from standing variation (e.g., Conte *et al.* [2012] *Proceedings of the National Academy of Sciences* 279[5039]; Lee & Coop [2017] *Genetics* 207[1591]). For instance, the classic example of freshwater stickleback evolving lateral plates from standing variation in *Eda* is commonly referred to as ‘parallel adaptation’ (e.g., Roesti *et al.* [2014] *Molecular Ecology* 23[16]). Moreover, Stern’s suggestion of establishing ‘collateral evolution’ to distinguish *de novo* mutation and standing variation as sources of repeatability, however useful, has not been widely embraced. Therefore we have retained our use of the term ‘parallel’ throughout, although in the Discussion we have offered our interpretation that standing variation is the likely source of the bulk of the parallel adaptation signals, with a reference to Stern’s review (lines 401-405).

This study mentioned that the herbarium samples could be traced back to 1830. Do these herbarium samples over a large time span? This manuscript does not describe the collection time of herbarium samples but uniformly classify them as historic. It should be clarified if all historic samples have the same collection time (the same year or a similar year). Otherwise, the authors should be cautious about biases in allele frequencies of each historical population. In addition, analysis results (e.g., historic vs. modern F_{ST} and θ_{PI}) related to allele frequencies in historic locations may be affected.

We thank the reviewer for this comment, which provided an opportunity to further clarify the presentation of our work. The collection years and locations for all samples (as well as which analyses they were used in) are now detailed in table S4. Historic samples were collected over a large span of time (1830-1973, with the more than 90% collected before 1940), however we have updated our historic-modern population comparisons to tighten the temporal range of historic samples used for each population (the largest collection year range for a historic population sample now twenty years), and removed populations where this was not possible (details: table S11). We provide a more detailed response to the question of bias in the historical/modern genome scans below (final major point of Reviewer 4). Furthermore, for the

analysis of temporal changes of the haploblocks, we have now included time as a continuous variable in the logistic regressions (lines 304-305; tables S12, S16, S17) to avoid pooling historic samples. Modeling time as continuous gave qualitatively similar results to the analysis in which we pooled the historic data at each geographic location, and both analyses are now presented in the Results.

The authors performed ABBA-BABA analysis using the same genus species and found that the source of variation in some haploblocks was introgression. But this result is no longer discussed. Adaptive introgression can be a driver of local adaptation. Then a question to the author is, could this affect their interpretation of their results?

We thank the reviewer for this creative perspective on our data, which presented an opportunity for us to prune our presented analyses to only the most crucial for the study. Although an interesting hypothesis, the original source of the haploblocks does not affect the interpretation of our main finding of recent climate mediate adaptation in Europe. This is the case because the haploblocks appear to be old (certainly predating the introduction of *A. artemisiifolia* to Europe), given how common and widespread they are in North America at both time points. As the ABBA-BABA analysis is not used in our Discussion, we have elected to entirely remove the analysis from the manuscript.

Line 435-437: It was confused that which version of BUSCO was employed in this study, as you mean 253 complete genes is 99.2% of the total? Besides, I think the percentage of the number of duplicated genes (45.1%) was suspicious for a high-quality genome assembly in general. Please check and describe detailly.

We used BUSCO version 5.3.1 and the eukaryota odb10 dataset, which we now state in the main text (lines 489-491). Paleopolyploidy is common in plants and is known to occur frequently in the Asteraceae (Barker *et al.* [2008] Molecular Biology and Evolution, 25[11]), and this likely contributes to the high fraction of duplicated genes in this assembly. Running this same analysis on the chromosome-level *Helianthus annuus* (from the same family and tribe as *A. artemisiifolia*) assembly yielded similar BUSCO scores, including a relatively high number of complete and duplicated BUSCO genes (115 duplicated genes in *A. artemisiifolia* and 85 duplicated genes in sunflower):

A. artemisiifolia:

C:99.2%[S:54.1%,D:45.1%],F:0%,M:0.8%,n:255

H. annuus:

C:99.2%[S:65.9%,D:33.3%],F:0.4%,M:0.4%,n:255

It is possible that a small number of highly differentiated regions in the maternal and paternal chromosomes are mistakenly retained as haplotigs in our haploid assembly. However, we do not observe large sections of scaffolds that have low read depths and low heterozygosity, when mapping the HiC reads back to the reference as single end (fig. S19). If sections of scaffolds were to contain haplotigs, then they should be homozygous and have approximately ½ the genome wide read depth on average. Very few of these regions (lower depth and heterozygosity) were identified on scaffolds that contain haploblocks, suggesting that if there are

any residual haplotigs in the haploid assembly, they are not interfering with the identification of the haploblocks, as the Lostruct analysis is run on each scaffold separately.

Line 418-426: As you mentioned that the hifiasm assembled heterozygous contigs in one set of genome sequence, and the Purge_dups was always used to remove duplicated regions. However, the validation only based on the reduction of duplicated BUSCO genes was not convincing. It is necessary to do more effort on the validation on the genome assembly as it influenced the accuracy of the subsequent analysis directly. Firstly, self-to-self contigs alignment should be made to check the redundancy of genome assembly in a dot-plot. It is also essential to check the sequence depth and the heterozygosity using Illumina reads as the heterozygous contigs tend to show higher sequence depth and heterozygosity. Tool like GAAP (<https://github.com/zy-optimistic/GAAP>) could be employed to verify the redundancy. Command as follow: “gaap snvcov -r genome.fasta -i ngs1.fastq -l ngs2.fastq -l tgs.fastq -t threads”.

We agree that heterozygosity and sequencing depth are approaches that can be used to identify redundancy. In fact, the purge_dups pipeline removes contigs using both heterozygosity and sequencing depth. This is the exact approach we used to remove the haplotigs and produce the final assembly.

In a good-faith effort to respond to the reviewer’s request for further validation, we now include the self-to-self alignment (fig. S20) of the final genome assembly, the dot-plot of which revealed little redundancy. Based on the above suggestion, we further assessed the presence of remnant haplotigs and other assembly artifacts by aligning the Illumina reads from the reference genome back to the assembled and haplotig-purged reference. We showed that sequencing depth (averaged across 1-Mbp windows) was never greater than two times higher, or 0.5 times lower, than the mean, and furthermore that regions of both low depth and low heterozygosity did not correspond to the haploblocks we identified. This analysis is now described in the manuscript in lines 505-517 and visualized in fig. S19.

Additional minor comments:

1. Line 36-37: "We show highly significant levels of parallelism between Europe and North America, with upwards of 11.7% of climate adaptation signals shared between ranges." This sentence is not clear. I couldn't find the result corresponding to "11.7% of climate adaptation signals" in the text. Please consider re-writing it.

This sentence was removed from the abstract.

2. Line 111: "alignments of over 600 modern and historic genome sequences". I found the usage of these samples in the manuscript to be unclear. From the Method section (Line 453), I know that the number of modern samples is 311. However, in the Result section, the authors performed genome scans for population allele frequencies among 284 modern samples (Line 127-129) and performed GWAS on 121 samples (Line 137). The usage of historic samples is even more ambiguous. Please describe them clearly.

Thank you for this useful suggestion. Samples used for particular analyses have now been clearly described in the materials and methods (lines 519-530) and in table S4.

3. Line 137: The author should give the full name of GWAS.

Thank you for noting this oversight. Genome-wide association study is now written out in full at first mention (line 143).

4. Line 142-143: "Candidate SNPs in ELF3 are restricted to high-latitude populations in both ranges." I noticed that the latitude ranges for North American and European are not equivalent. Due to the climatic characteristics of the two continents, should the authors discuss the temperature difference between the two ranges at the same latitude?

We thank the reviewer for this excellent point. In ragweed's ranges, European locations tend to be warmer and wetter for a given latitude than North American populations (van Boheemen, Atwater & Hodgins [2019] *New Phytologist* 222[1]). Therefore despite the difference in latitude between the North American and European locations, the climatic conditions are very similar. We have stated this in-text (lines 151-153), and shown it in a supplementary figure (fig. S3). We also note that latitude is strongly correlated with the major axis of climate variation in both ranges (the slopes of PC1 and latitude are identical between the ranges although the intercepts are different; fig. S3), and so we decided to use latitude, instead of a PC1 of climate variables in the haploblock analysis since it is more transferable across studies as it doesn't change with the particular variables/populations included in the analysis.

In addition, it would be better to give references for ELF3.

We have included a reference for the functional characterization of *ELF3* in *A. thaliana* when the gene is first mentioned (lines 148-149).

5. Line 168: Does the legend at the bottom of Figure 1B refer to the number of samples in the population? The authors should change the breaks to integers.

That's correct. We've clarified the legend name and changed the breaks to integers in figure 1B.

6. Line 207: "... populations within a 200km radius of Berlin...". Please describe the number of these populations and list the ELF3 frequencies. It would be better to mark "200km radius of Berlin" in Figure 1B.

The 200km radius around Berlin captures two modern populations ($n = 8$; $n = 3$) and 15 historic samples. The use of "populations" was not really accurate here, and we have rephrased it as "across all samples within a 200km radius of Berlin (15 historic and eleven modern samples)" (lines 239-241).

7. Line 223: The x-axis of Figure 2B lacks ticks and tick labels. It would be better to clearly show the size and position of the HB448 on the scaffold448.

Figure 2B shows the entire scaffold; axis ticks and labels are now shown on the x-axis.

8. Line 240-241: Please list the name and size of the seven haploblocks.

Haploblock names and sizes are now stated in-text (lines 278-280).

9. Line 245-249: I found the results of introgression and recombination rates unclear. Please replace "some" with the exact number (Line 246). "Six haploblocks showed evidence of...". It would be better to enumerate the six haploblocks or the one with the exception.

We have removed the introgression analysis from the manuscript. We have also included a more detailed description of the recombination rate variation (lines 267-274).

12. Line 406: please add the citation of hifiasm.

Citation added (line 466).

13. Line 670-678: There seems puzzling to choose such a large window size (10 Mb). The authors should explain the reason.

We have removed the introgression analysis from the manuscript. The large windows were used because ABBA-BABA tests are known to perform poorly in small windows and the haploblock are of a large size, facilitating the use of large windows.

Reviewer #3 (Remarks to the Author):

Overview

The authors assembled a draft genome of *Ambrosia trifida* from a field population. They re-sequenced collections from north America and Europe for a GWAS in an attempt to see genomics regions under selection in both the native and invaded range. They identified one such locus that housed a gene that encodes for flowering time (ELF3) whose allele frequency changing with latitude. They were also able to identify several other loci under selection that were unique and/or shared between the two groups. Using historic and modern collections, they also identified a selective sweep possible large inversion in one of their scaffolds indicating a haploblock influencing local population structure. Globally, they were able to identify several other haploblocks whose alleles frequencies did (or did not) correlate with latitude. Some of these blocks varied the same in historic collections while others diverged.

General Comments

Overall I think the writing is excellent, extremely detailed and the work impressive. However, I think the story is inflated and the message is muddled by jargon and lots of intermingled data references. I wish the introduction more clearly stated what it is they are trying to test and whether it was sufficiently answered. I think the story is interesting and relevant but lacks clarity.

We thank the reviewer for this call for clarity, which we have heeded. Toward this end, we modified the Introduction to clarify the aims and questions of the study (lines 108-123). Further, we expanded the Discussion and tried to more clearly explain how our results directly address these aims (e.g., lines 421-423).

Furthermore, I think a lot is missing in the methods about what samples were taken where and when. It is hard to know exactly how to interpret the data without a clear idea of how many samples were being taken from each location (NA vs Europe) and modern versus herbarium.

We fully agree with the reviewer that these necessary details were accidentally excluded from the original version of the manuscript. We have updated the Methods to include a section outlining the samples used in each analysis (lines 519-530), and this is supported by a supplementary table (table S4).

Again, I think the authors did an amazing job with data analysis and collection but did not translate exactly what it all means, what was exciting about the work, and how they came to those conclusions. One last note, I think it is critical that your assembly and associated annotation files be made available upon publication. Many authors are merely depositing raw reads on NCBI to obfuscate their work. Please make sure to clearly state what data is going to be available to others.

We thank the reviewer for these kind words, and are delighted to hear that our work has been well received. We have no intentions to obfuscate our analyses or diminish the reproducibility of our work. We have now uploaded the genome assembly and annotation to figshare and the

assembly to NCBI. We have included links and accession numbers for this information in the data and code availability statement (lines 854-859).

Line Comments

Lines 58-62: There is a lot of absolutes in here. For instance, while it is often the case that an invasive species has a broad native range or that it stress resilient, it is not always. Some species have narrow ranges until they escape predation in a new environment, sometimes they are able to invade new ranges precisely because they have “comfortable” conditions and are not particularly resilient. I caution against saying things like “Invasive species also inhabit broad and climatically diverse ranges” because you are saying this has to be true in order for an invasion to be successful, which it is not.

We agree with the reviewer. Although range size and habitat breadth are two major dimensions of invasiveness, abundance is considered to be the third and an important component of invasion (by some definitions; Fristoe *et al.* [2021] *Proceedings of the National Academy of Sciences*, 118[22]). Although abundance is also correlated with habitat breadth and range size, it is not necessarily a requirement to meet several definitions of the broad and sometimes contentious term of invasive. Thus in response to this criticism, we modified the Introduction and Discussion to be less emphatic (lines 47-62; 394-405).

Lines 83-84: “High connectivity”? Do you mean high gene flow or that their patches are contiguous? This term is not clear to me.

We meant high gene flow; this has been changed in the manuscript (lines 77-78).

Ln124-135: But how different is Europe from North America in terms of latitude, precipitation, temperature, etc? I would not consider one more extreme then the other and imagine that much of the climate is the same in both the native and invaded region. It seems that you are constantly alluding to an idea that the invaded range is “Extreme” but it is not. For instance, If a plant is well adapted to invading disturbed soybean fields in the us, the same is going to be true in Europe, Asian, and Africa.

Although some studies have identified a niche shift in Europe compared to North America (e.g., Sun *et al.* [2017] *Ecosphere* 8[4]), climatic conditions at locations where European ragweed populations were sampled are within the extremes of the North American population samples (we have demonstrated this in a supplementary figure [fig. S3] and have mentioned this in the Discussion [lines 398-399]). Nevertheless, introductions from North America are unlikely to have been a good genetic match for the individual European location to which they were initially introduced, suggesting parallelism between Europe and North America is the result of rapid local adaptation *within* the European range since the species' establishment on the continent. Furthermore, it is this parallelism in climatic gradients identified between the ranges that likely contributes to the observed parallelism in the local adaptation candidates between the ranges.

Figure 1C: Why are the scales on the y axis for North America and Europe drastically different?

More extreme XtX values (analogous to F_{ST} but with correction for population structure) in North America when compared with Europe reflects greater population differentiation in North America, which is to be expected in the native vs. invasive range and reflects the greater amount of time since the establishment of the native range populations. We had previously mentioned this but have now expounded upon it (lines 167-170).

Ln 182: A shift in what direction? Earlier in the year?

The change in probability of flowering over time depended on the latitude and the day of collection. For flowering, the northern accessions were more likely to be flowering in later collection years compared to earlier collection years. For fruiting and flowering, plants from later collection years were less likely to be flowering/fruiting earlier in the season and more likely to be flowering/fruiting later in the season. We added a further explanation of this to help clarify the results (lines 211-213). We note that our goal was to identify if phenological traits were impacted by collection year, and latitude while controlling for the time of year the plant was collected, as an indicator that these traits have changed over time and space in Europe. Our goal was not to predict the local flowering and fruiting times or estimate genetic clines in these traits over space and time using these data. This would be a fascinating avenue for future research but would require more granular phenological stages (which may not be possible in ragweed given its floral architecture), local growing season information, and more complex statistical modeling (as per Wu and Colautti [2022] *Proceeding of the National Academy of Sciences* 119[18]).

Ln 185-186: “more recently collected plants were more likely to be flowering than older historic specimens.” Were these collected at approximately the same time of year then? its not exactly clear.

As these were historic samples collected by botanists over many decades (since 1830), we could not control the collection times. However, in the statistical analysis we controlled for the time of year the samples were collected (Julian day of the year). Collection time had a large effect on the probability of flowering and fruiting. However, we still identified a significant impact of the year of collection at high latitudes (year*latitude interaction $P=0.0051$; table S10; fig. S5) even when controlling for the time of year the samples were collected. We lengthened the description of these results to clarify this (lines 203-221).

Ln 202: What does this rapid adaptation depend on? I imagine for rapid adaptation to occur, the introduced plants must have sufficient diversity for adaptation to occur. What is genetic diversity like in the invaded and native range for this species (I believe it is primarily out crossing?) What exactly is it maladapted to? What in the invaded range is selecting these plants so hard since I imagine precipitation, light, and temperature in continental Europe to be similar to southern Canada and northern US.

Ragweed has not been observed to self-fertilize, and levels of within population genetic diversity between ranges is similar (Bieker *et al.* [in press] *Science Advances*). While Europe is climatically similar to North America, there is substantial variation in climate within ragweed's range on both continents. Multiple studies have clearly demonstrated *local adaptation* to climate within each range using trait and marker data (e.g., Sun *et al.* [2017] *Ecosphere* 8(4); van Boheeman & Hodgins [2020] *Molecular Ecology* 29[21]; McGoey, Hodgins & Stinchcombe

[2020] *Ecology and Evolution* 10[11]) Therefore, haphazard introduction of North American plants to various environments in Europe would likely result in maladaptation to these local European climates, particularly given that the European invasion is primarily sourced from a single North American admixture cluster located in the eastern part of the American midwest (Bieker *et al.* [in press] *Science Advances*).

Ln 351: Why would it be striking? Assuming that the same genetics were in both the invaded and native range and the climate in Mainland Europe and North America are relatively similar, would you not expect similar patterns and convergent evolution?

We find it striking that these signatures of parallelism have arisen so quickly; we have adjusted this section to clarify this (lines 399-401). Further there are a growing number of studies that do not find high levels of parallelism in the genetic basis of adaptation, even when comparing within a single species (e.g., Elmer *et al.* [2014] *Nature Communications* 5[5168]; Gould & Stinchcombe [2017] *Molecular Ecology* 26[1]; Szukala *et al.* [2022] *Molecular Ecology*). Therefore we think it is important to highlight that significant levels of parallel evolution can occur at the genetic level in some cases, as it might point to the factors that drive convergent and divergent evolutionary change at the genetic level.

Ln 362: Couldn't inversions be verified in your various plants by looking at read-pair breaking in your resequencing data? It is relatively easy to confirm inversions using low-coverage resequencing with either PacBio or Paired end Illumina.

Our resequencing data used Illumina reads, and the utility of these reads to identify inversions can be confounded by repetitive regions at inversion breakpoints (e.g. Sharakhov *et al.* [2006] *Proceedings of the National Academy of Sciences* 103[16]). This was also the case with the sunflower inversions described in Todesco *et al.* (2020) *Nature* 584(7822) (Greg Owens pers. comm.), and appears to be the case in ragweed also. We tried identifying breakpoints with both short reads and long reads using the reference individual. Although we believe our reference individual was heterozygous for one of the putative inversions, the regions around the breakpoints are highly repetitive and this prevented both assembly and alignment based approaches for the inversions using long and short reads.

Ln 381: You use the word plasticity but I am not sure if you are referring to phenotypic plasticity, genotypic plasticity, genetic diversity or some combination of these.

We meant phenotypic plasticity, and this is now clarified in-text (line 443).

Ln 437: I would like to see a better summary of the genome assembly and many more statistics to help evaluate its completeness. Though I trust you I would like to know more about the predicted genome size, number of chromosomes and how close you are to achieving this. It is unclear from the limited information in the supplementary tables.

We have now included a more detailed analysis of the genome assembly and several more statistics to help evaluate its completeness and quality (lines 125-140; 452-517). We note the predicted genome size based on flow cytometry and expected chromosome number provided in the text (lines 129-130).

First we estimated the genome size and ploidy from the reads using GenomeScope 2. This confirmed that the sample was diploid and also revealed that the sample was both highly heterozygous and provided a genome size estimate similar to the expectations based on flow cytometry (lines 478-484; fig. S16). These results were also similar to our genome assembly size and L90, which we further highlighted in our results (lines 129-130).

Second, based on Reviewer 2's suggestion, we realigned our short reads to the reference and examined read depth and heterozygosity along the length of each scaffold in windows. We did this to determine if large haplotigs were still prevalent in our assembly, as these regions would have low heterozygosity and low read depth. We did not identify large haplotig blocks, and none that would have interfered with our downstream analysis (e.g., if large haplotigs were on the same scaffold as our haploblocks these could have contributed to their false identification through a local PCA based analysis of outlier regions). We also looked for regions with high read depth and high heterozygosity as these regions could have been false collapsed duplicated regions. However, we identified no evidence of this (lines 505-513; fig. S19).

Ln391: I am shocked there isn't an inclusion of which samples were collected from where and when. Maybe this is in the supplemental but understanding the time and shape of the populations collected will help with the overall evaluation of evolution. This is especially true if samples from the herbarium predate the use of herbicides to control this plant as I can imagine modern agriculture would apply many novel stresses that supersede climate. Furthermore, it is unclear from the methods if the samples were taken from open space, agriculture, and/or some other areas. Again, this may influence the downstream analysis but it is unclear.

We agree with all the reviewers that this was a major oversight, which has been rectified in the present version of the manuscript. We have updated the Methods to include a section outlining the samples used in each analysis (lines 519-530), supported by a supplementary table (table S4). We have also included the rationale for selecting samples in specific methods/results sections.

We appreciate the reviewer's interest in agricultural stressors. We note that the reviewer mistakenly referred to the focal species as *A. trifida* (giant ragweed). This species is a larger agricultural pest in North America than the focal species of this paper, *A. artemisiifolia* (common ragweed). However, agricultural stressors, of course, are of interest to us as well. Based on our analysis thus far, climate based selection is a main driver of trait variation across the landscape and is the main cause of adaptive genetic differentiation among populations within the native and introduced ranges. This is supported by the fact that most XtX outliers are also associated with environmental variables. Common ragweed populations, including the ones we sampled, are found in disturbed roadside environments or abandoned lots and fields. Some of our populations are adjacent to agricultural fields, but few were sampled from within fields and none were from known herbicide resistant populations. In a separate study we may look specifically for evidence of selection on herbicide resistance genes, but given the predominance of climate

mediated selection and the importance of this for pollen production, climate change induced range shifts and for human health, we decided to focus on this topic.

Reviewer #4 (Remarks to the Author):

This study investigates local adaptation in the invasive weed *Ambrosia artemisiifolia*. A new, good quality reference genome was assembled for this species of worldwide importance. An impressive set of over 600 samples were sequenced. This material includes contemporary populations as well as historical material from herbarium and covers both the native range (North America) and the main invasive range (Europe). Various methods including phenotype scoring from herbarium images, genome-wide association mapping and different genome scan analyses were implemented to detect genomic regions associated with local adaptation.

Analyses were very thoroughly conducted and revealed parallel changes in the native and invasive ranges as well as congruence between the spatial and temporal patterns of genetic changes at a few genomic regions, including some large haploblocks. This convincingly demonstrates that a few genomic regions with large effects have contributed to local adaptation. Further, the temporal sampling scheme demonstrated that local adaptation was ancient in the native range, whereas in Europe, initial maladaptation was followed by rapid adaptation to local climates.

I have only few concerns about this study. My main concern is about the comparison of historic and contemporary populations.

First, the plant material is not described in details. Readers are referred to reference #10, which is not satisfactory, as this is not a published paper but a bioRxiv deposit. More details are needed about the sampling locations, dates of sampling and number of individuals per population (for both herbarium and contemporary populations). This could be provided as a supplementary material.

This major oversight was identified by all reviewers, and has been rectified. We have updated the Methods to include a section outlining the samples used in each analysis (lines 519-530), supported by a supplementary table (table S4). We have also included the rationale for selecting samples in specific methods/results sections, and a further table summarizing the details of the historic and modern populations compared (table S11).

The comparison of historic vs modern population was based on F_{st} , on the ratio of historic to modern nucleotide diversity, and on the modelling of allele frequency shift in time, for pairs of historic and modern populations said to have been sampled at the same locations. This approach seems to rely on the hypothesis that the modern populations that were sampled descended from the historic populations retrieved from herbarium, and that rapid adaptation has occurred in the time interval between the two samplings. This scenario implies that no extensive migration nor population replacement have taken place. This seems a precarious hypothesis for an invasive plant that thrives in perturbed environments.

Please refer to the response to this question below.

The authors should give more details regarding their criteria for selecting the 16 pairs of populations considered.

Selection criteria are described on lines 225-227, and data on these populations are now provided in table S4 and table S11.

They possibly had some historical records in favour of the stability of populations over time?

According to results presented in Bieker *et al.* (in press) *Science Advances*, in terms of genetic composition, it appears populations in North America have remained relatively stable while populations in Europe have varied in their stability. However, Bieker *et al.* have also shown that genome-wide F_{ST} between modern and historic populations is exceedingly low both in the native (North American modern-historic $F_{ST} < 0.006$ in all cases except the southern cluster) and in the introduced range (Europe modern-historic $F_{ST} = 0.006$).

Also, what are the time intervals between historical and modern populations? Are these intervals similar for all the considered pairs?

We have regrouped historic populations and excluded samples to minimize the variation in historic sampling time within populations, however the time periods of historical samples do vary between populations. This is unavoidable, because we were highly constrained by the samples available in herbaria. In Europe, this is also strongly impacted by the range expansion itself. Some regions were invaded much longer ago (e.g., France) than others (e.g., eastern Europe). Years of sampling of individuals and populations are now presented in table S4 and table S11. We have also removed components of our pooled analyses of sweep candidates across populations (e.g. gene ontology enrichment) due to the variation in time periods between populations. We repeated our spatio-temporal analysis of haploblock frequency using time as a continuous variable and found almost identical results compared to the analysis where there were two discrete time-points (modern-historic; table S12). Furthermore, our analysis of HB448, which contains the majority of the sweep signals in Europe, showed substantial changes in haplotype frequency using these logistic regressions.

In the other hand, it may well be that historic populations were composed of haphazardly introduced, maladapted genotypes that were subsequently replaced by more adapted genotypes, so that there is no ancestry relationship between paired historical and modern samplings. I think that this scenario will also leave a signature consistent with a selective sweep, and probably a signature of a strong selective sweep in the case of population extinction followed by replacement. However, in my mind, this process of demographic turn-over and sorting is different from rapid adaptation via selective sweep, and should be distinguished as such.

We believe that migration could have contributed to differences in some of the populations we scanned, and this is evident in the variation in the distribution of F_{ST} values in different populations (see y-axes in fig. S6-S7), although we also note that these F_{ST} values are similar in both the native and introduced ranges despite the introduced range experiencing evidence of

population structure change over time (Bieker *et al.* [in press] *Science Advances*). However, our scans for sweeps are relative to the overall differences between each historic and modern population, and therefore even in a population which has experienced a large genetic turnover, our outliers show more extreme shifts relative to the genome-wide pattern. Additionally, we believe our requirement for an extreme reduction in genetic diversity corresponding to extreme F_{ST} in a region relative to the genome wide pattern will further exclude artifacts of high turnover in populations, since substantially higher diversity in the historic samples must be present, thus indicating that the putatively selected haplotype is likely segregating in the past, but rises to high frequencies in the present, thereby reducing modern variation. Finally, we focussed on loci that showed both spatial and temporal signatures of selection and particularly those that showed parallel patterns of spatial selection in North America and Europe. False positives due to purely neutral processes (e.g., population replacement in the absence of selection) are therefore less likely to have occurred as we are not relying on the identification of outliers from this selective sweep analysis alone for candidate identification. Local adaptation in an invasion scenario with multiple introductions is likely a continuum from spatial sorting and demographic turnover to classic local sweep scenarios, but with highly outcrossing species like ragweed a scenario in the middle seems most likely. Ragweed develops an extensive seedbank, is wind pollinated and highly outcrossing, and wholesale demographic turnover (especially in the absence of selection) in specific geographic locations seems unlikely. This is further supported by the low F_{ST} values between modern and historic populations in Europe and North America (Bieker *et al.* [in press] *Science Advances*). However, given the extensive genetic parallelism we identified, pre-existing standing variation must be critical for this plant's rapid adaptation in Europe. Indeed the regions we identified as the major contributors to the temporal sweep signals were HB448 and *ELF3*. Both variants of HB448 and *ELF3* are present in the native range and also show signatures of spatial selection in these regions. We are not claiming these variants evolved *de novo* through a classic sweep in Europe (both selected variants are present in the historic European samples and in North America) but rather that the source of these variants in the introduced range was the native range variants. Further analysis of these specific regions reveals that changes in candidate alleles over time showed extreme shifts in frequency relative to the genome-wide patterns. In fact compared to the null SNPs, few if any variants showed an equal amount of change. We feel that this is strong evidence that these regions are under selection over time, even if migration contributed to the source of the adaptive variation.

Other comments

I.151. I don't quite understand the rationale behind combining consecutive windows to account for linkage disequilibrium (more a personal thought, this does not necessarily necessitate any change)

It is an assumption of the weighted-Z analysis that analysis windows are independent, an assumption which may be violated in cases of extensive linkage disequilibrium, and as such inflate any estimates of parallelism. To account for this possibility we combined consecutive windows and windows within haploblocks (which could be non-independent as a result of

extended linkage disequilibrium) into single windows and repeated the parallelism test. We have reworded our description of this (lines 177-181).

I. 245-247. The authors implemented ABBA-BABA tests using either *Ambrosia trifida* or *Ambrosia psilostachya* as putative donor species and found some evidence for introgression. This result is presently very shortly and not discussed. It is not at all essential for the study and could be omitted. On the other hand, it raises some questions: to my knowledge, hybridization between *A. artemisiifolia* and either *A. trifida* or *A. psilostachya* has not been described in the literature. *A. psilostachya* is reproducing almost exclusively vegetatively, and, if I'm correct, its distribution does not overlap with that of *A. artemisiifolia*. Hybridization between these species seems rather unlikely and I wonder whether this could be incomplete lineage sorting rather than introgression.

We thank the reviewer for this lucid observation. We agree that the introgression analysis is not essential to our study, thus we have entirely removed the analysis from the manuscript. Incidentally, hybrids of *A. artemisiifolia* and *A. trifida* are well known, and *A. artemisiifolia* and *A. psilostachya* do overlap in range (Vincent & Cappadocia [1987] *Weed Science* 35[5]; Wagner & Beals [1958] *Rhodora* 60[715]). In addition, all three species were introduced to Europe around the same time, thus potentially allowing hybridization in the introduced range. A recent paper has clearly identified signals of hybridization and introgression among these species (Bieker *et al.* [in press] *Science Advances*).

L. 248 From Figure S10, it seems that evidence for a reduced recombination rates across haploblocks varies much depending on the genetic map considered. Hence, it is not clear to me, how the authors could conclude that six of the haploblocks showed reduced recombination consistent with inversions.

That recombination reductions are map-specific is exactly the expectation if a structural variant were causing the reduction in recombination. If a parent is heterozygous for an inversion, recombination will be suppressed. If a parent is homozygous for an inversion, recombination will not be suppressed in the region. Unfortunately, the GBS markers do not allow us to determine if the parents were heterozygous for the haploblock. However, we can conclude that we do not see either general suppression of recombination in the haploblock regions, or high levels of recombination in each haploblock region in all crosses, which is consistent with an inversion. We have improved our explanation of these results to help clarify the meaning of these findings (lines 269-274).

L. 252-253 “we genotyped all historic and modern samples by local, divergent population structure in haploblock regions” I find this part of the sentence quite unclear, and maybe the authors should rephrase in a more explanatory way.

We have reworded this section to add clarity (lines 262-274).

From the material and methods, I understood that the genotype of each sample was assigned from k-means clustering into three groups (the two homozygotes and the heterozygotes), using the first two components of a PCA. From fig S7 it seems that

separation into three groups is not always clear-cut. This suggests that there was quite a lot of remnant genetic variation with each collapsed “genotype”. I find it hard to assess whether ignoring this underlying genetic variation may have biased the tests of statistical association with latitude or time in one way or the other. Do the authors have any view about this?

PCA plots include (low-coverage) historic samples, which account for at least some of the noise. We have updated our haplotype calling in the historic samples by only considering loci with divergent allele frequencies between the homozygotes in the modern samples. Using these diagnostic SNPs improved our capacity to identify the three clusters and reduced the ambiguity. We should note that it is likely, given the variation found within both haplotypes for these haploblocks, that these haploblocks are relatively old and they contain variation within the haploblock that is under selection irrespective of the haploblock itself. For example, the early flower allele in *ELF3* is only found in one background of HB2. Indeed we find it likely that since these haploblocks contain an over-representation of certain types of genes with functions important for traits under selection (e.g. flower time genes) that mutations segregating within them will frequently be under selection. However, we were interested in determining if genetic differences in association with the haploblock itself were under selection. This is why we examined the changes in haploblock frequency over time and their associations with traits. As well as the analyses which rely on haploblock genotypes, we have also performed many tests utilizing individual SNPs (e.g. BayPass, GWAS, selective sweep scans) which support many of the associations with time and latitude seen in the haploblocks. In fact, for each haploblock the diagnostic SNPs show a significant over-representation of XtX outlier SNPs, which is what we might expect if selection on the haploblock itself is a main contributor to the signals of selection in these regions.

I. 314 and I. 331: This is rather speculative but I wondered whether selection for enhanced dichogamy, or selection for traits that favour allogamy could have taken place during invasion of Europe? This could have favoured admixture, generating genetic variation useful for local adaptation.

After improving our haploblock genotyping (see above), this particular association was no longer significant. More generally, we are not confident in our statistical power to differentiate overdominance from dominance, and have therefore removed the overdominance tests from the manuscript.

Material and Methods

L. 439 and following: Sample alignment, variant calling

The genome re-sequencing method, obtained read coverage and approach for variant calling (seems to be a probabilistic approach using *angds*) are not explicitly stated.

For analyses using only modern samples, we used SNPs called with GATK and included information regarding the read depth and other variant filtering metrics in our descriptions (detailed in “Sample alignment, variant calling and filtering”, lines 532-549). ANGSD was used for analyses incorporating historic and modern samples (as historic samples are sequenced to a

lower depth on average) and its use is described in the methods for these analyses (line 668, 685, 735).

Reviewers' Comments:

Reviewer #1:

Remarks to the Author:

The authors have addressed my concerns. Only one suggestion is that it may be more appropriate to delete "near chromosome-level" or change it to "high quality".

Reviewer #2:

Remarks to the Author:

This re-submitted manuscript is much more precise than the previous submission. The authors have addressed all my concerns and clarified some key concepts and methods. However, I think the quality of the genome assembly still needs to be improved. The authors cite an article (SA paper) in the main text and responses, known as the Biorivx article (ref 11). Moreover, the resequencing data used in this study came from that SA paper. Although the genomes assembled in the two manuscripts are from different individuals, the genome assembled in this article is not much improved compared to that in the SA article. When the authors responded to Reviewer1's third point, they mentioned working on a phased diploid assembly. Therefore, we strongly recommend that the authors describe the phased diploid assembly in this study and use it as a reference genome for genotyping and haploblock identification. I believe this will significantly improve the quality of the article.

Other comments

I don't think the author's answer to the fourth question of Reviewer4 is correct. Reviewer4 considered that the comparison of population pairs should stand on the hypothesis that the modern population that was sampled descended from the historical population. I agree with Reviewer4's point. Based on this hypothesis, temporal selection scans are only needed to compare five North American and seven European pairs. Otherwise, if the modern Berlin samples are partly descended from the France population, some new comparison pairs occur (e.g., modern Berlin vs. historical France). Overall, the authors should carefully consider the effect of neutral processes. I suggest that the authors elucidate the dynamic histories of the 12 population pairs (modern and historical). Then, do sequence simulations to evaluate this selective sweep scan strategy based on the population history. In addition, I suggest that the authors use some existing tests for the selective sweep, such as the Tajima's D test and the Fay & Wu's H test.

Line 129. "an L90 of 20 (A. artemisiifolia has 18 chromosomes)". I suggest the authors list the length and name of the 20 scaffolds in TableS.

Line 246. The horizontal axis labels of Figure 2 are not corrected as described in the rebuttal.

Line 279. The authors have followed the previous comments and listed the sizes of the seven haploblocks. I suggest listing the scaffolds on which the seven haploblocks located.

Reviewer #3:

Remarks to the Author:

The authors assembled a draft genome of *Ambrosia trifida* from a field population. They re-sequenced collections from north America and Europe for a GWAS in an attempt to see genomics regions under selection in both the native and invaded range. They identified one such locus that housed a gene that encodes for flowering time (ELF3) whose allele frequency changing with latitude. They were also able to identify several other loci under selection that were unique and/or shared between the two groups. Using historic and modern collections, they also identified a selective sweep possible large inversion in one of their scaffolds indicating a haploblock influencing local population structure. Globally, they were

able to identify several other haploblocks whose alleles frequencies did (or did not) correlate with latitude. Some of these blocks varied the same in historic collections while others diverged.

In this revision, the authors addressed nearly all major concerns and worked hard to clarify the story, add much needed detail about the historical sampling, details about the genome assembly, and clarified the message and major findings of the research. The work is considerable and relatively novel in the field of weed genomics. I feel like their considerate response to the reviewers comments and efforts to update the manuscript helped considerably. My major concerns are addressed and I am happy to see this manuscript move forward.

REVIEWER COMMENTS

Reviewer #1 (Remarks to the Author):

The authors have addressed my concerns. Only one suggestion is that it may be more appropriate to delete "near chromosome-level" or change it to "high quality".

We thank the reviewer for their comments and are pleased to know that we have addressed them sufficiently. As we have improved our reference genome, we now refer to it as "chromosome-level phased diploid" (line 127).

Reviewer #2 (Remarks to the Author):

This re-submitted manuscript is much more precise than the previous submission. The authors have addressed all my concerns and clarified some key concepts and methods. However, I think the quality of the genome assembly still needs to be improved. The authors cite an article (SA paper) in the main text and responses, known as the Biorivx article (ref 11). Moreover, the resequencing data used in this study came from that SA paper. Although the genomes assembled in the two manuscripts are from different individuals, the genome assembled in this article is not much improved compared to that in the SA article. When the authors responded to Reviewer1's third point, they mentioned working on a phased diploid assembly. Therefore, we strongly recommend that the authors describe the phased diploid assembly in this study and use it as a reference genome for genotyping and haploblock identification. I believe this will significantly improve the quality of the article.

We thank the reviewer for their previous round of comments and are glad to hear that they were sufficiently addressed. We strongly disagree that the Science Advances article compromised the novelty of this study. The reference genome was not intended to be the focus of either study, and although the studies share resequencing data, the analyses conducted with them are entirely distinct. The Science Advances study examined the population structure and invasion history as well as the microbial community of ragweed. It did not examine adaptation to local climates, structural variation and parallel patterns of adaptation which is the focus of this study.

Importantly we have shown that these findings are significant and novel. However we are grateful to the reviewer for pushing us to include our phased diploid assembly in this revised version, which has (as they suggested) dramatically improved our manuscript by providing clear support that at least 26% of the haploblocks are indeed inversions. We note that our present genome is assembled to the chromosome level and is haplotype-resolved, with each set of haploid chromosomes containing ~95% of the haploid genome and a complete set of BUSCO genes. This represents a vast improvement compared to the Science Advances study genome, which was highly fragmented with each chromosome represented by many thousands of relatively small contigs (N50=89.7kbp).

Other comments

I don't think the author's answer to the fourth question of Reviewer4 is correct. Reviewer4 considered that the comparison of population pairs should stand on the hypothesis that the modern population that was sampled descended from the historical population. I agree with Reviewer4's point. Based on this hypothesis, temporal selection scans are only needed to compare five North American and seven European pairs. Otherwise, if the modern Berlin samples are partly descended from the France population, some new comparison pairs occur (e.g., modern Berlin vs. historical France). Overall, the authors should carefully consider the effect of neutral processes. I suggest that the authors elucidate the dynamic histories of the 12 population pairs (modern and historical). Then, do sequence simulations to evaluate this selective sweep scan strategy based on the population history. In addition, I suggest that the authors use some existing tests for the selective sweep, such as the Tajima's D test and the Fay & Wu's H test.

We thank the author for this suggestion. We conducted Fay & Wu's *H* test of selection in each modern population that also had paired historic samples. We discovered strong evidence of selective sweeps using this test. In Europe, these putatively selected regions closely matched the sweeps identified in the historic-modern comparisons for hb-chr2 and ELF3 in Berlin. Interestingly, sweep signals were geographically more widespread for hb-chr2 using this test, and included evidence of selection in multiple European populations and a few North American populations, suggesting the sweep may pre-date some of our historic samples. We certainly do not rule out the possibility of gene flow from the native range contributing to the rapid evolutionary change in the north of the European range. However, the multiple tests of selection (GEA, Fay & Wu's *H*, XtX), the links of those loci to adaptive traits such as flowering time, and

the parallel patterns of allele frequency change with the environment as well as trait evolution provide strong support for the hypothesis that selection contributed to these allele frequency shifts across Europe.

Line 129. "an L90 of 20 (A. artemisiifolia has 18 chromosomes)". I suggest the authors list the length and name of the 20 scaffolds in TableS.

This is a good suggestion. We have listed the chromosomes and their sizes in each haploid assembly in a new supplementary table, table S2.

Line 246. The horizontal axis labels of Figure 2 are not corrected as described in the rebuttal.

We thank the reviewer for spotting and reporting this mistake. The x-axis is now annotated in what is now figure 3B.

Line 279. The authors have followed the previous comments and listed the sizes of the seven haploblocks. I suggest listing the scaffolds on which the seven haploblocks located.

The names of the haploblocks refer to the scaffold on which they occur; e.g., HB448 was a haploblock on scaffold 448 in the previous version. To clarify this, we have amended our naming system throughout the manuscript. hb-chr2 now refers to the haploblock on chromosome 2.

Reviewer #3 (Remarks to the Author):

The authors assembled a draft genome of *Ambrosia trifida* from a field population. They re-sequenced collections from north America and Europe for a GWAS in an attempt to see genomics regions under selection in both the native and invaded range. They identified one such locus that housed a gene that encodes for flowering time (ELF3) whose allele frequency changing with latitude. They were also able to identify several other loci under selection that were unique and/or shared between the two groups. Using

historic and modern collections, they also identified a selective sweep possible large inversion in one of their scaffolds indicating a haploblock influencing local population structure. Globally, they were able to identify several other haploblocks whose alleles frequencies did (or did not) correlate with latitude. Some of these blocks varied the same in historic collections while others diverged.

In this revision, the authors addressed nearly all major concerns and worked hard to clarify the story, add much needed detail about the historical sampling, details about the genome assembly, and clarified the message and major findings of the research. The work is considerable and relatively novel in the field of weed genomics. I feel like their considerate response to the reviewers comments and efforts to update the manuscript helped considerably. My major concerns are addressed and I am happy to see this manuscript move forward.

We thank the reviewer for their suggestions, and are happy to know that their concerns have been addressed.

Reviewers' Comments:

Reviewer #2:

Remarks to the Author:

In this revision, the authors addressed my concerns. They updated the genome assembly to chromosome-level phased diploid assembly and added selective sweep tests to detect positive selection. I am happy to see that four of the large haploblocks showed inversion in diploid reference genome, and I suggest they place the figS12 in the main text.

REVIEWERS' COMMENTS

Reviewer #2 (Remarks to the Author):

In this revision, the authors addressed my concerns. They updated the genome assembly to chromosome-level phased diploid assembly and added selective sweep tests to detect positive selection. I am happy to see that four of the large haploblocks showed inversion in diploid reference genome, and I suggest they place the figS12 in the main text.

We thank reviewer #2 for their comments and are pleased to know that they have been sufficiently addressed. We have moved Fig. S12 to the main text; it is now Fig. 4.